# NetPyNE, a tool for data-driven multiscale modeling of brain circuits

Salvador Dura-Bernal[1]*, Benjamin A Suter[2†], Padraig Gleeson[3], Matteo Cantarelli[4], Adrian Quintana[5], Facundo Rodriguez[1,4], David J Kedziora[6], George L Chadderdon[1‡], Cliff C Kerr[6], Samuel A Neymotin[1,7], Robert A McDougal[8,9], Michael Hines[8], Gordon MG Shepherd[2], William W Lytton[1,10]

[1]Department of Physiology & Pharmacology, State University of New York Downstate Medical Center, Brooklyn, United States; [2]Department of Physiology, Northwestern University, Chicago, United States; [3]Department of Neuroscience, Physiology and Pharmacology, University College London, London, United Kingdom; [4]MetaCell LLC, Boston, United States; [5]EyeSeeTea Ltd, Cheltenham, United Kingdom; [6]Complex Systems Group, School of Physics, University of Sydney, Sydney, Australia; [7]Nathan Kline Institute for Psychiatric Research, Orangeburg, United States; [8]Department of Neuroscience and School of Medicine, Yale University, New Haven, United States; [9]Center for Medical Informatics, Yale University, New Haven, United States; [10]Department of Neurology, Kings County Hospital, Brooklyn, United States

*For correspondence: salvadordura@gmail.com

Present address: †Institute of Science and Technology (IST) Austria, Klosterneuburg, Austria; ‡Burnet Institute, Melbourne, Australia

**Abstract** Biophysical modeling of neuronal networks helps to integrate and interpret rapidly growing and disparate experimental datasets at multiple scales. The NetPyNE tool (www.netpyne. org) provides both programmatic and graphical interfaces to develop data-driven multiscale network models in NEURON. NetPyNE clearly separates model parameters from implementation code. Users provide specifications at a high level via a standardized declarative language, for example connectivity rules, to create millions of cell-to-cell connections. NetPyNE then enables users to generate the NEURON network, run efficiently parallelized simulations, optimize and explore network parameters through automated batch runs, and use built-in functions for visualization and analysis – connectivity matrices, voltage traces, spike raster plots, local field potentials, and information theoretic measures. NetPyNE also facilitates model sharing by exporting and importing standardized formats (NeuroML and SONATA). NetPyNE is already being used to teach computational neuroscience students and by modelers to investigate brain regions and phenomena.
DOI: https://doi.org/10.7554/eLife.44494.001

## Introduction

The worldwide upsurge of neuroscience research through the BRAIN Initiative, Human Brain Project, and other efforts is yielding unprecedented levels of experimental findings from many different species, brain regions, scales and techniques. As highlighted in the BRAIN Initiative 2025 report (*Bargmann et al., 2014*), these initiatives require computational tools to consolidate and interpret the data, and translate isolated findings into an understanding of brain function (*Shou et al., 2015*; *Fisher et al., 2013*). Biophysically detailed multiscale modeling (MSM) provides a promising approach for integrating, organizing and bridging many types of data. Individual experiments often are limited to a single scale or level: for example, spiking activity in vivo, subcellular connectivity in

**eLife digest** The approximately 100 billion neurons in our brain are responsible for everything we do and experience. Experiments aimed at discovering how these cells encode and process information generate vast amounts of data. These data span multiple scales, from interactions between individual molecules to coordinated waves of electrical activity that spread across the entire brain surface. To understand how the brain works, we must combine and make sense of these diverse types of information.

Computational modeling provides one way of doing this. Using equations, we can calculate the chemical and electrical changes that take place in neurons. We can then build models of neurons and neural circuits that reproduce the patterns of activity seen in experiments. Exploring these models can provide insights into how the brain itself works. Several software tools are available to simulate neural circuits, but none provide an easy way of incorporating data that span different scales, from molecules to cells to networks. Moreover, most of the models require familiarity with computer programming.

Dura-Bernal et al. have now developed a new software tool called NetPyNE, which allows users without programming expertise to build sophisticated models of brain circuits. It features a user-friendly interface for defining the properties of the model at molecular, cellular and circuit scales. It also provides an easy and automated method to identify the properties of the model that enable it to reproduce experimental data. Finally, NetPyNE makes it possible to run the model on supercomputers and offers a variety of ways to visualize and analyze the resulting output. Users can save the model and output in standardized formats, making them accessible to as many people as possible.

Researchers in labs across the world have used NetPyNE to study different brain regions, phenomena and diseases. The software also features in courses that introduce students to neurobiology and computational modeling. NetPyNE can help to interpret isolated experimental findings, and also makes it easier to explore interactions between brain activity at different scales. This will enable researchers to decipher how the brain encodes and processes information, and ultimately could make it easier to understand and treat brain disorders.

DOI: https://doi.org/10.7554/eLife.44494.002

brain slices, and molecular processes in dissociated or cultured tissue. These data domains cannot be compared directly, but can be potentially integrated through multiscale simulations that permit one to switch readily back-and-forth between slice-simulation and in vivo simulation. Furthermore, these multiscale models permit one to develop hypotheses about how biological mechanisms underlie brain function. The MSM approach is essential to understand how subcellular, cellular and circuit-level components of complex neural systems interact to yield neural function or dysfunction and behavior (*Markram et al., 2015*; *Skinner, 2012*; *MindScope et al., 2016*). It also provides the bridge to more compact theoretical domains, such as low-dimensional dynamics, analytic modeling and information theory (*Churchland and Sejnowski, 2016*; *Churchland and Abbott, 2016*; *Cunningham and Yu, 2014*).

NEURON is the leading simulator in the domain of multiscale neuronal modeling (*Tikidji-Hamburyan et al., 2017*). It has 648 models available via ModelDB (*McDougal et al., 2017*), and over 2000 NEURON-based publications (https://neuron.yale.edu/neuron/publications/neuron-bibliography). However, building data-driven large-scale networks and running parallel simulations in NEURON is technically challenging (*Lytton et al., 2016*), requiring integration of custom frameworks to build and organize complex model components across multiple scales. Other key elements of the modeling workflow such as ensuring replicability, optimizing parameters and analyzing results also need to be implemented separately by each user (*Mulugeta et al., 2018*; *McDougal et al., 2016*). Lack of model standardization makes it difficult to understand, reproduce and reuse many existing models and simulation results.

We introduce a new software tool, NetPyNE (Networks using Python and NEURON). NetPyNE addresses these issues and relieves the user from much of the time-consuming programming previously needed for these ancillary modeling tasks, automating many network modeling requirements

for the setup, run, explore and analysis stages. NetPyNE enables users to consolidate complex experimental data with prior models and other external data sources at different scales into a unified computational model. Users can then simulate and analyze the model in the NetPyNE framework in order to better understand brain structure, brain dynamics and ultimately brain structure-function relationships. The NetPyNE framework provides: (1) flexible, rule-based, high-level standardized specifications covering scales from molecule to cell to network; (2) efficient parallel simulation both on stand-alone computers and in high-performance computing (HPC) clusters; (3) automated data analysis and visualization (e.g. connectivity, neural activity, information theoretic analysis); (4) standardized input/output formats, importing of existing NEURON cell models, and conversion to/from NeuroML (*Gleeson et al., 2010*; *Cannon et al., 2014*); (5) automated parameter tuning across multiples scales (molecular to network) using grid search and evolutionary algorithms. All tool features are available programmatically or via an integrated graphical user interface (GUI). This centralized organization gives the user the ability to interact readily with the various components (for building, simulating, optimizing and analyzing networks), without requiring additional installation, setup, training and format conversion across multiple tools.

NetPyNE's high-level specifications are implemented as a declarative language designed to facilitate the definition of data-driven multiscale network models by accommodating many of the intricacies of experimental data, such as complex subcellular mechanisms, the distribution of synapses across fully detailed dendrites, and time-varying stimulation. Contrasting with the obscurity of raw-code descriptions used in many existing models (*McDougal et al., 2016*), NetPyNE's standardized language provides transparent and manageable descriptions. These features in particular promise to increase the reproducibility of simulation results and the reuse of models across research groups. Model specifications are then translated into the necessary NEURON components via built-in algorithms. This approach cleanly separates model specifications from the underlying technical implementation. Users avoid complex low-level coding, preventing implementation errors, inefficiencies and flawed results that are common during the development of complex multiscale models. Crucially, users retain control of the model design choices, including the conceptual model, level of biological detail, scales to include, and biological parameter values. The NetPyNE tool allows users to shift their time, effort and focus from low-level coding to designing a model that matches the biological details at the chosen scales.

NetPyNE is one of several tools that facilitate network modeling with NEURON: neuroConstruct (*Gleeson et al., 2007*), PyNN (*Davison, 2008*), Topographica (*Bednar, 2009*), ARACHNE (*Aleksin et al., 2017*) and BioNet (*Gratiy et al., 2018*). NetPyNE differs from these in terms of the range of scales, from molecular up to large networks and extracellular space simulation – it is the only tool that supports NEURON's Reaction-Diffusion (RxD) module (*McDougal et al., 2013*; *Newton et al., 2018*). It also provides an easy declarative format for the definition of complex, experimentally derived rules to distribute synapses across dendrites. NetPyNE is also unique in integrating a standardized declarative language, automated parameter optimization and a GUI designed to work across all these scales.

NetPyNE therefore streamlines the modeling workflow, consequently accelerating the iteration between modeling and experiment. By reducing programming challenges, our tool also makes multiscale modeling highly accessible to a wide range of users in the neuroscience community. NetPyNE is publicly available from http://netpyne.org, which includes installation instructions, documentation, tutorials, example models and Q&A forums. The tool has already been used by over 50 researchers in 24 labs to train students and to model a variety of brain regions and phenomena (see http://netpyne.org/models) (*Dura-Bernal et al., 2018*; *Romaro et al., 2018*; *Lytton et al., 2017*; *Neymotin et al., 2016b*). Additionally, it has been integrated with other tools in the neuroscience community: the Human Neocortical Neurosolver (https://hnn.brown.edu/) (*Jones et al., 2009*; *Neymotin et al., 2018*), Open Source Brain (http://opensourcebrain.org) (*Gleeson et al., 2018*; *Cannon et al., 2014*), and the Neuroscience Gateway portal (http://nsgportal.org) (*Sivagnanam et al., 2013*).

## Results

### Tool overview and workflow

NetPyNE's workflow consists of four main stages: (1) high-level specification, (2) network instantiation, (3) simulation and (4) analysis and saving (*Figure 1*). The first stage involves defining all the parameters required to build the network, from population sizes to cell properties to connectivity rules, and the simulation options, including duration, integration step, variables to record, *etc*. This is the main step requiring input from the user, who can provide these inputs either programmatically with NetPyNE's declarative language, or by using the GUI. NetPyNE also enables importing of existing cell models for use in a network.

The next stages can be accomplished with a single function call – or mouse click if using the GUI. The network instantiation step consists of creating all the cells, connections and stimuli based on the high-level parameters and rules provided by the user. The instantiated network is represented as a Python hierarchical structure that includes all the NEURON objects required to run a parallel simulation. This is followed by the simulation stage, where NetPyNE takes care of distributing the cells and

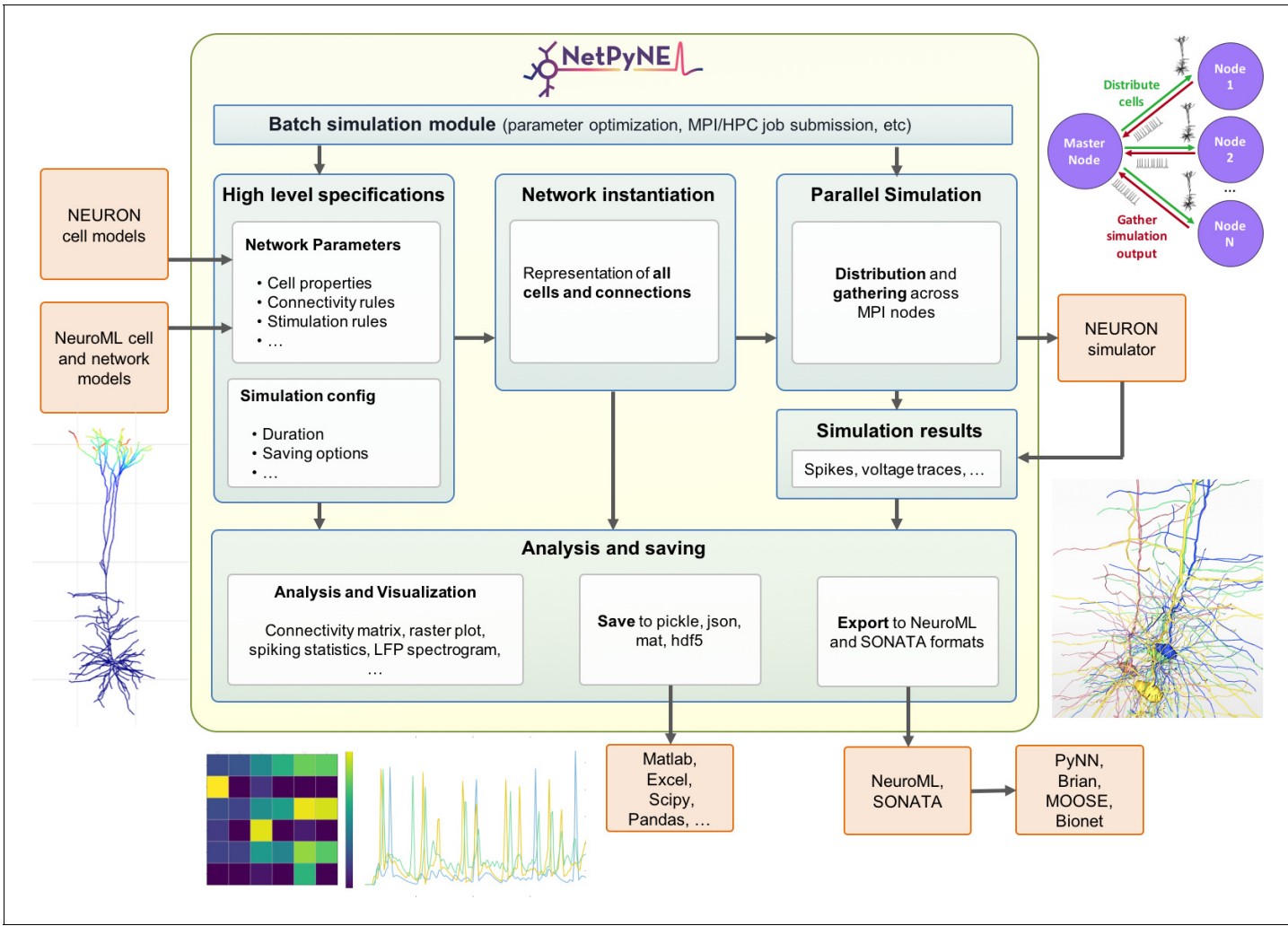

**Figure 1.** Overview of NetPyNE components and workflow. Users start by specifying the network parameters and simulation configuration using a high-level JSON-like format. Existing NEURON and NeuroML models can be imported. Next, a NEURON network model is instantiated based on these specifications. This model can be simulated in parallel using NEURON as the underlying simulation engine. Simulation results are gathered in the master node. Finally, the user can analyze the network and simulation results using a variety of plots; save to multiple formats or export to NeuroML. The Batch Simulation module enables automating this process to run multiple simulations on HPCs and explore a range of parameter values.
DOI: https://doi.org/10.7554/eLife.44494.003

connections across the available nodes, running the parallelized simulation, and gathering the data back in the master node. Here, NetPyNE is using NEURON as its back-end simulator, but all the technical complexities of parallel NEURON are hidden from the user. In the final stage, the user can plot a wide variety of figures to analyze the network and simulation output. The model and simulation output can be saved to common file formats and exported to NeuroML, a standard description for neural models (*Cannon et al., 2014*). This enables exploring the data using other tools (e.g. MATLAB) or importing and running the model using other simulators (e.g. NEST).

An additional overarching component enables users to automate these steps to run batches of simulations to explore model parameters. The user can define the range of values to explore for each parameter and customize one of the pre-defined configuration templates to automatically submit all the simulation jobs on multi-processor machines or supercomputers.

Each of these stages is implemented in modular fashion to make it possible to follow different workflows such as saving an instantiated network and then loading and running simulations at a later time. The following sections provide additional details about each simulation stage.

## High-level specifications

A major challenge in building models is combining the data from many scales. In this respect, NetPyNE offers a substantial advantage by employing a human-readable, clean, rule-based shareable declarative language to specify networks and simulation configuration. These standardized high-level specifications employ a compact JSON-compatible format consisting of Python lists and dictionaries (*Figure 2*). The objective of the high-level declarative language is to allow users to accurately describe the particulars and patterns observed at each biological scale, while hiding all the complex technical aspects required to implement them in NEURON. For example, one can define a probabilistic connectivity rule between two populations, instead of creating potentially millions of cell-to-cell connections with Python or hoc for loops. The high-level language enables structured specification of all the model parameters: populations, cell properties, connectivity, input stimulation and simulation configuration.

### Population and cell parameters

Users define network populations, including their cell type, number of cells or density (in $cells/mm^3$), and their spatial distribution. *Figure 2A–i,ii* show setting of *yrange* and alternatively setting *numCells* or *density* for two cell types in the network. Morphological and biophysical properties can then be applied to subsets of cells using custom rules. This enables, for example, setting properties for all cells in a population with a certain 'cell type' attribute or within a spatial region. The flexibility of the declarative rule-based method allows the heterogeneity of cell populations observed experimentally to be captured. It also allows the use of cell implementations of different complexity to coexist in the same network, useful in very large models where full multi-scale is desired but cannot be implemented across all cells due to the computational size of the network. These alternative implementations could include highly simplified cell models such as Izhikevich, Adaptive Exponential Integrate-and-Fire (AdEx) or pre-calculated point neuron models (*Lytton and Stewart, 2006*; *Naud et al., 2008*; *Izhikevich, 2003*). These can be combined in the same network model or swapped in and out: for example (1) explore overall network dynamics using simple point-neuron models; (2) re-explore with more biologically realistic complex models to determine how complex cell dynamics contribute to network dynamics. We also note that order of declaration is arbitrary; as here, one can define the density of typed cells before defining these types. In *Figure 2A–iii,iv*, we define the two different *PYR* models whose distribution was defined in A-i,ii. The *simple* model is simple enough to be fully defined in NetPyNE – one compartment with Hodgkin-Huxley (*hh*) kinetics with the parameters listed (here the original *hh* parameters are given; typically these would be changed). More complex cells could also be defined in NetPyNE in this same way. More commonly, complex cells would be imported from hoc templates, Python classes or NeuroML templates, as shown in *Figure 2A-iv*. Thus, any cell model available online can be downloaded and used as part of a network model (non-NEURON cell models must first be translated into NMODL/Python) (*Hines and Carnevale, 2000*). Note that unlike the other statements, *Figure 2A-iv* is a procedure call rather than the setting of a dictionary value. The `importCellParams()` procedure call creates a new dictionary with

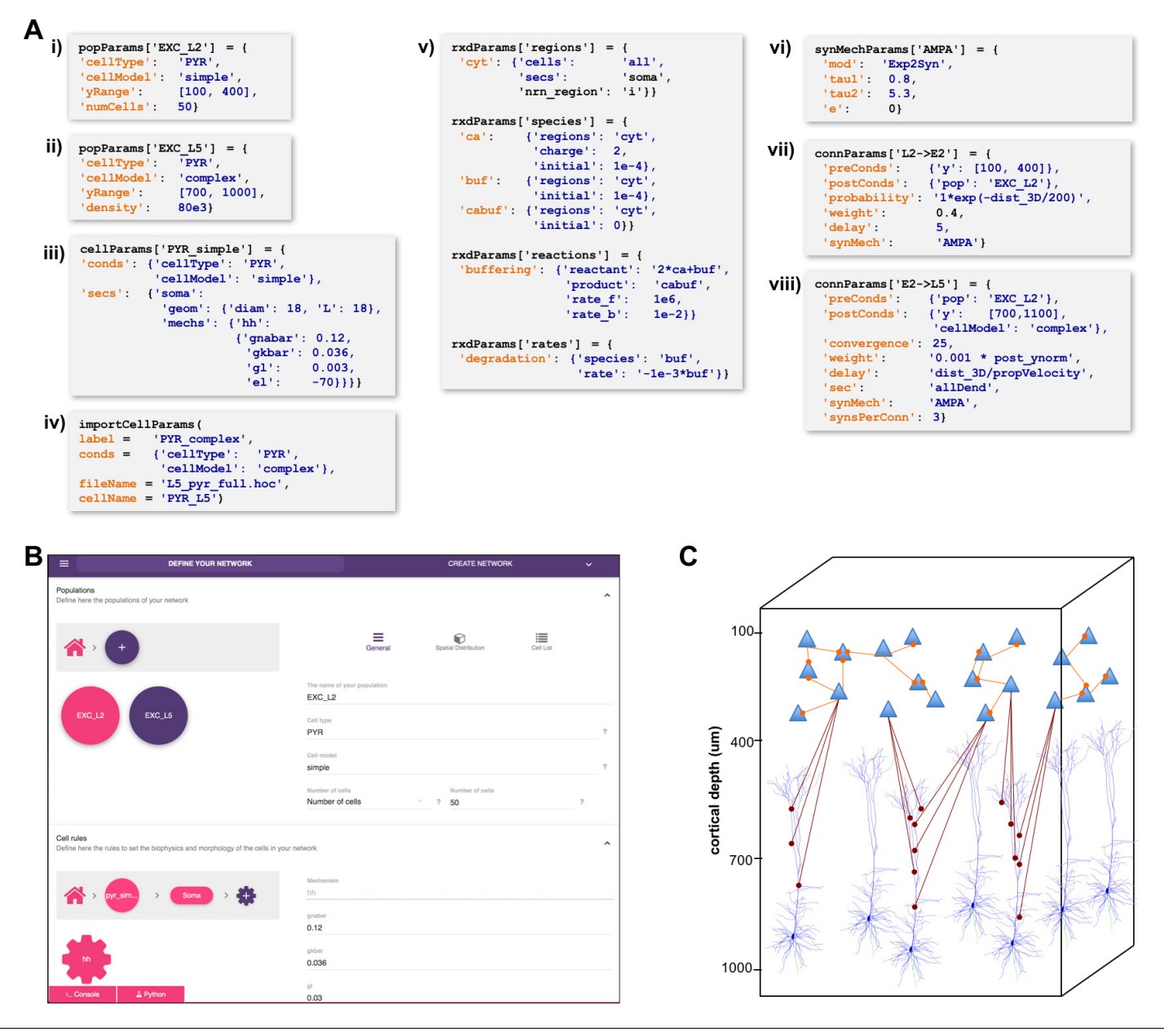

**Figure 2.** High-level specification of network parameters. (**A**) Programmatic parameter specification using standardized declarative JSON-like format. i, ii: specification of two populations iii,iv: cell parameters; v: reaction-diffusion parameters; vi,vii,viii: synapse parameters and connectivity rules. (**B**) GUI-based parameter specification, showing the definition of populations equivalent to those in panel A. (**C**) Schematic of network model resulting from the specifications in A.

DOI: https://doi.org/10.7554/eLife.44494.004

NetPyNE's data structure, which can then be modified later in the script or via GUI, before network instantiation.

## Reaction-diffusion parameters

NetPyNE's declarative language also supports NEURON's reaction-diffusion RxD specifications of Regions, Species, States, Reactions and Rates (https://neuron.yale.edu/neuron/docs/reaction-diffusion) (*McDougal et al., 2013*; *Newton et al., 2018*). RxD simplifies the declaration of the chemophysiology – intracellular and extracellular signaling dynamics – that complements electrophysiology.

During network instantiation, RxD declarative specifications are translated into RxD components within or between cells of the NetPyNE-defined network. This adds additional scales – subcellular, organelle, extracellular matrix – to the exploration of multiscale interactions, for example calcium regulation of hyperpolarization-activated cyclic nucleotide–gated (HCN) channels promoting persistent network activity (*Neymotin et al., 2016a*; *Angulo et al., 2017*). RxD is now being extended to also permit definition of voltage-dependent or voltage- and ligand-dependent ion channels, and can also interact with NMODL-defined mechanisms so as to respond to synaptic events and affect membrane voltage.

RxD specifications in NetPyNE are organized using a logical sequence of questions: (1) where do dynamics occur?, (2) who are the actors?, (3) what are the reactions? This sequence, and the syntax, are similar to direct use of RxD, except that NetPyNE uses a declarative language consisting of nested dictionaries with strings and values, instead of directly instantiating the Python. The example in *Figure 2A–v* implements a simplified model of calcium buffering with a degradable buffer: $2Ca + Buf \leftrightarrow CaBuf, Buf \rightarrow (degraded)$. Calcium dynamics, including buffering, play a major role in neuronal plasticity and firing activity (*Blackwell, 2013*; *Bhalla, 2017*). In the example, we first indicate in the `rxdParams['regions']` dictionary *where* the dynamics will occur: in the cytosol of the soma of all cells (`cyt`). NetPyNE facilitates this step by allowing the user to select all or a subset of cells by population name, relative index and/or cell global ids. Next, we specify *who* are the actors involved via `rxdParams['regions']`: free calcium ions (`cyt`), free buffers (`buf`) and calcium-bound buffers (`cabuf`). Finally, we define *what* reactions will occur using the `rxdParams['reactions']` and `rxdParams['rates']` dictionaries: calcium buffering and buffer degradation. These RxD mechanisms will dynamically affect the cytosolic concentration of calcium (`cai`), a shared variable that can also be read and modified by NMODL-defined ionic channels and synaptic mechanisms. This establishes all interactions among RxD, NMODL, and NEURON-currents, coupling reaction-diffusion dynamics to cell and network electrophysiology.

To exemplify how RxD components can affect network dynamics, we implemented a more elaborate demonstration model linking the concentration of inositol triphosphate (IP3) to network activity. The model consisted of a three-layer cortical network of five-compartment neurons with multiple NMODL-based mechanisms, including sodium, potassium, calcium and HCN channels. We added an RxD system of intracellular neuronal calcium and IP3 signaling in all compartments of all neurons. Cytosolic and endoplasmic reticulum (ER) regions were represented by fractional volume. ER included IP3 receptors (IP3Rs) with a slow calcium inactivation binding site, sarco/ER Ca2+-ATP-ase (SERCA) pumps, and calcium leak. Ion concentrations in the 3D extracellular space surrounding the network were also modeled. The model demonstrated multiscale dynamics from molecular to network scales, showing how metabotropic activation (not explicitly modeled but represented as an increase in cytosolic IP3) would influence local field potential (LFP). Ignoring the influence of the recurrent dynamics at each scale, we could trace influences in the following sequence: increased cytosol IP3 → ER IP3R activation → ER calcium released to cytosol → activation of $Ca^{2+}$-dependent $K^+$ channels → hyperpolarization → reduced network firing → reduced LFP. The code and further details of this example are available at https://github.com/Neurosim-lab/netpyne/tree/development/examples/rxd_net (copy archived at https://github.com/elifesciences-publications/netpyne/tree/development/examples/rxd_net).

## Connectivity and stimulation parameters

NetPyNE is designed to facilitate network design. Connectivity rules are flexible and broad in order to permit ready translation of many different kinds of experimental observations. Different subsets of pre- and post-synaptic cells can be selected based on a combination of attributes such as cell type and spatial location (*Figure 2A–v,vi*). Users can then specify one or multiple target synaptic mechanisms (e.g. AMPA, AMPA/NMDA or GABA_A). In the case of multicompartment cells, synapses can be distributed across a list of cell locations. Multiple connectivity functions are available including all-to-all, probabilistic, fixed convergence and fixed divergence. The connectivity pattern can also be defined by the user via a custom connectivity matrix. Additionally, several connectivity parameters, including probability, convergence weight and delay, can be specified as a function of pre- and post-synaptic properties, using arbitrarily defined mathematical expressions. This permits instantiation of biological correlations such as the dependence of connection delay on distance, or a

fall-off in connection probability with distance. Electrical gap junctions and learning mechanisms – including spike-timing dependent plasticity and reinforcement learning – can also be incorporated.

NetPyNE supports specification of subcellular synaptic distribution along dendrites. This allows synaptic density maps obtained via optogenetic techniques to be directly incorporated in networks. *Figure 3A* left shows the layout for one such technique known as sCRACM (subcellular Channelrho-dopsin-2-Assisted Circuit Mapping) (*Petreanu et al., 2009*). A density map of cell activation mea-sured from the soma is determined by photostimulating a brain slice containing channelrhodopsin-tagged pre-synaptic boutons from a defined source region (in this example, from the thalamus; *Figure 3A*). NetPyNE randomly distributes synapses based on location correspondence on a den-dritic tree which can be either simple or multicompartmental (*Figure 3B*). Here again, the automa-tion of synapse placements permits models of different complexity to be readily swapped in and out. Depending on the data type and whether one wants to use averaging, the location maps may be based on 1D, 2D, or 3D tissue coordinates, with the major *y*-axis reflecting normalized cortical depth (NCD) from pia to white matter. Alternatively, NetPyNE can define synapse distributions based on categorical information for dendritic subsets: for example obliques, or spine densities, or on path distance from the soma, apical nexus or other point. As with the density maps, these rules will automatically adapt to simplified morphologies. NetPyNE permits visualization of these various synaptic-distribution choices and cellular models via dendrite-based synapse density plots (*Figure 3C*), which in this case extrapolates from the experimental spatial-based density plot in *Figure 3A* (*Hooks et al., 2013*; *Petreanu et al., 2009*; *Suter and Shepherd, 2015*).

Network models often employ artificial stimulation to reproduce the effect of afferent inputs that are not explicitly modeled, for example ascending inputs from thalamus and descending from V2 tar-geting a V1 network. NetPyNE supports a variety of stimulation sources, including current clamps, random currents, random spike generators or band-delimited spike or current generators. These can be placed on target cells using the same flexible, customizable rules previously described for con-nections. Users can also employ experimentally recorded input patterns.

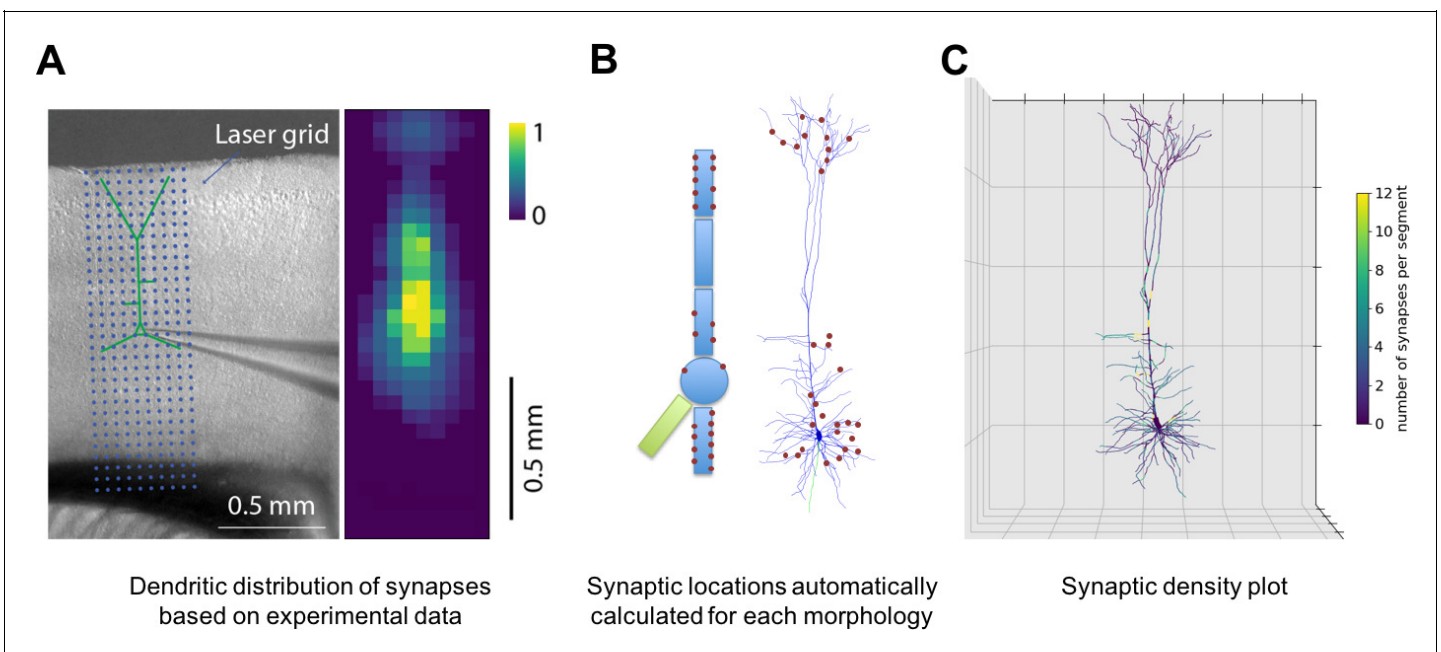

**Figure 3.** Specification of dendritic distribution of synapses. (A) Optogenetic data provides synapse density across the 2D grid shown at left (*Suter and Shepherd, 2015*). (B) Data are imported directly into NetPyNE which automatically calculates synapse location in simplified or full multicompartmental representations of a pyramidal cell. (C) Corresponding synaptic density plot generated by NetPyNE.
DOI: https://doi.org/10.7554/eLife.44494.005

## Simulation configuration

Thus far, we have described the data structures that define network parameters: popParams, cellParams, connParams, etc. Next, the user will configure parameters related to a particular simulation run, such as simulation duration, time-step, parallelization options, *etc*. These parameters will also control output: which variables to plot or to record for graphing – for example, voltage or calcium concentration from particular cells, LFP recording options, file save options, and in what format, *etc*. In contrast to network and cell parameterization, all simulation options have default values so only those being customized are required.

## Network instantiation

NetPyNE generates a simulatable NEURON model containing all the elements and properties described by the user in the rule-based high-level specifications. As described above, declarations may include molecular processes, cells, connections, stimulators and simulation options. After instantiation, the data structures of both the original high-level specifications and the resultant network instance can be accessed programmatically or via GUI.

Traditionally, it has been up to the user to provide an easy way to access the components of a NEURON network model, for example the connections or stimulators targeting a cell, the sections in a cell, or the properties and mechanisms in each section. This feature is absent in many existing models. Hence, inspecting s models requires calling multiple NEURON functions (e.g. `SectionList.allroots()`, `SectionList.wholetree()` and `section.psection()`). Other models include some form of indexing for the elements at some scales, but since this is not enforced, their structure and naming can vary significantly across models.

In contrast, all networks generated by NetPyNE are consistently represented as a nested Python structure. The root of the instantiated network is the *net* object (*Figure 4*). *net* contains a list of cells; each cell contains lists or dictionaries with its properties, sections, and stimulators. Each section *sec* contains dictionaries with its morphology and mechanisms. For example, once the network is instantiated, the sodium conductance parameter for cell #5 can be accessed as `net.cells[5].secs.soma.mechs.hh.gbar`. This data structure also includes all the NEURON objects – Sections, NetCons, NetStims, IClamps, etc. embedded hierarchically, and accessible via the `hObj` dictionary key of each element.

## Parallel simulation

Computational needs for running much larger and more complex neural simulations are constantly increasing as researchers attempt to reproduce fast-growing experimental datasets (*Bezaire et al., 2016*; *Markram et al., 2015*; *MindScope et al., 2016*; *Dura-Bernal et al., 2018*; *Hereld et al., 2005*; *Lytton et al., 2016*). Fortunately, parallelization methods and high-performance computing (HPC, supercomputing) resources are becoming increasingly available to the average user (*Hines, 2011*; *Hines et al., 2008*; *Migliore et al., 2006*; *Towns et al., 2014*; *Amunts et al., 2016*; *Sivagnanam et al., 2013*; *Krause and Thörnig, 2018*).

The NEURON simulator provides a *ParallelContext* module, which enables parallelizing the simulation computations across different nodes. However, this remains a complex process that involves distributing computations across nodes in a balanced manner, gathering and reassembling simulation results for post-processing, and ensuring simulation results are replicable and independent of the number of processors used. Therefore, appropriate and efficient parallelization of network simulations requires design, implementation and deployment of a variety of techniques, some complex, many obscure, mostly inaccessible to the average user (*Lytton et al., 2016*).

NetPyNE manages these burdensome tasks so that the user can run parallelized simulations with a single function call or mouse click. Cells are distributed across processors using a round-robin algorithm, which generally results in balanced computation load on each processor (*Migliore et al., 2006*; *Lytton et al., 2016*). After the simulation has run, NetPyNE gathers in the master node all the network metadata (cells, connections, etc.) and simulation results (spike times, voltage traces, LFP signal, etc.) for analysis. As models scale up, it becomes impractical to store the simulation results on a single centralized master node. NetPyNE offers distributed data saving methods that reduce both the runtime memory required and the gathering time. Distributed data saving allows multiple

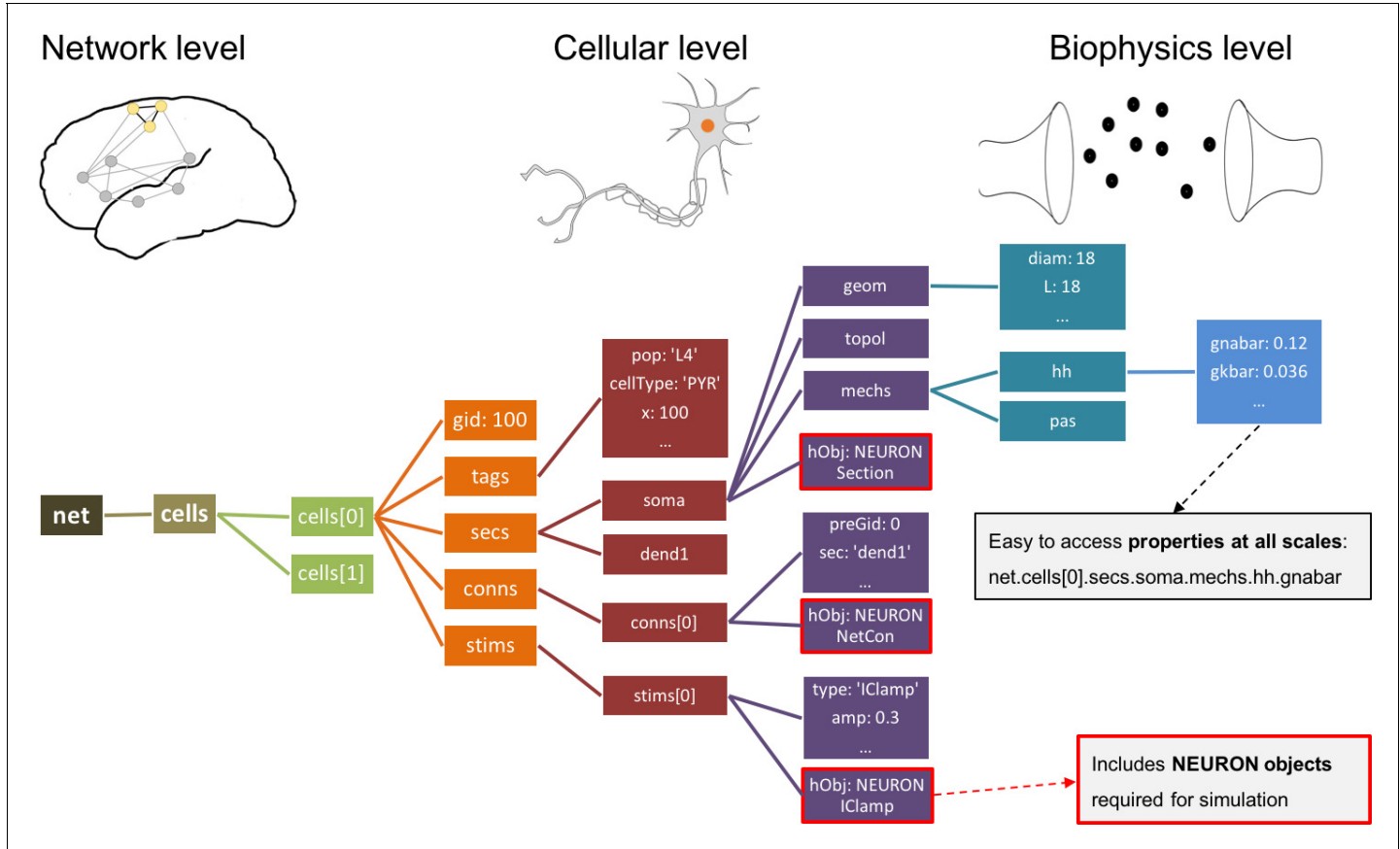

**Figure 4.** Instantiated network hierarchical data model. The instantiated network is represented using a standardized hierarchically organized Python structure generated from NetPyNE 's high-level specifications. This data structure provides direct access to all elements, state variables and parameters to be simulated. Defined NEURON simulator objects (represented as boxes with red borders) are included within the Python data structure.
DOI: https://doi.org/10.7554/eLife.44494.006

compute nodes to write information in parallel, either at intervals during simulation runtime, or once the simulation is completed. The output files are later merged for analysis.

Random number generators (RNGs) are often problematic in hand-written parallelized code; careful management of seeds is required since even use of the same seed or seed-sets across nodes will result in different random streams when the number of nodes is changed. Since random values are used to generate cell locations, connectivity properties, spike times of driving inputs, etc., inconsistent streams will cause a simulation to produce different results when switching from serial to parallel or when changing the number of nodes. In NetPyNE, RNGs are initialized based on seed values created from associated pre- and post-synaptic cell global identifiers (gids) which ensures consistent results across different numbers of cores. Specific RNG streams are associated to *purposive* seeds (e.g. connectivity or locations) and to a global seed, allowing different random, but replicable, networks to be run by modifying the single global seed. Similarly, manipulation of *purposive* seeds can be used to run, for example, a network with identical wiring but different random driving inputs.

We previously performed parallelization performance analyses, demonstrating that run time scales appropriately as a function of number of cells (tested up to 100,000) and compute nodes (tested up to 512) (**Lytton et al., 2016**). Simulations were developed and executed using NetPyNE and NEURON on the XSEDE Comet supercomputer via the Neuroscience Gateway (**Sivagnanam et al., 2013**). The Neuroscience Gateway, which provides neuroscientists with free and easy access to supercomputers, includes NetPyNE as one of the tools available via their web portal. Larger-scale models – including the M1 model with 10 thousand multicompartment neurons and 30 million synapses (**Dura-Bernal et al., 2018**) and the thalamocortical model with over 80 thousand point neurons and 300 million synapses (**Potjans and Diesmann, 2014**; **Romaro et al., 2018**) – have

been simulated on both the XSEDE Comet supercomputer and Google Cloud supercomputers. Run time to simulate 1 second of the multicompartment-neuron network required 47 minutes on 48 cores, and 4 minutes on 128 cores for the point-neuron network.

## Analysis of network and simulation output

To extract conclusions from neural simulations it is necessary to use further tools to process and present the large amounts of raw data generated. NetPyNE includes built-in implementations of a wide range of visualization and analysis functions commonly used in neuroscience (*Figure 5*). All analysis functions include options to customize the desired output. Functions to visualize and analyze network structure are available without a simulation run: (1) intracellular and extracellular RxD species concentration in a 2D region; (2) matrix or stacked bar plot of connectivity; (3) 2D graph representation of cell locations and connections; and (4) 3D cell morphology with color-coded variable (e.g. number of synapses per segment). After a simulation run, one can visualize and analyze simulation output: (1) time-resolved traces of any recorded cell variable (e.g. voltage, synaptic current or ion concentration); (2) relative and absolute amplitudes of post-synaptic potentials; statistics (boxplot) of spiking rate, the interspike interval coefficient of variation (ISI CV) and synchrony (*Kreuz et al., 2015*); power spectral density of firing rates; and information theoretic measures, including normalized transfer entropy and Granger causality.

A major feature of our tool is the ability to place extracellular electrodes to record LFPs at any arbitrary 3D locations within the network, similar to the approach offered by the LFPy (*Lindén et al.,*

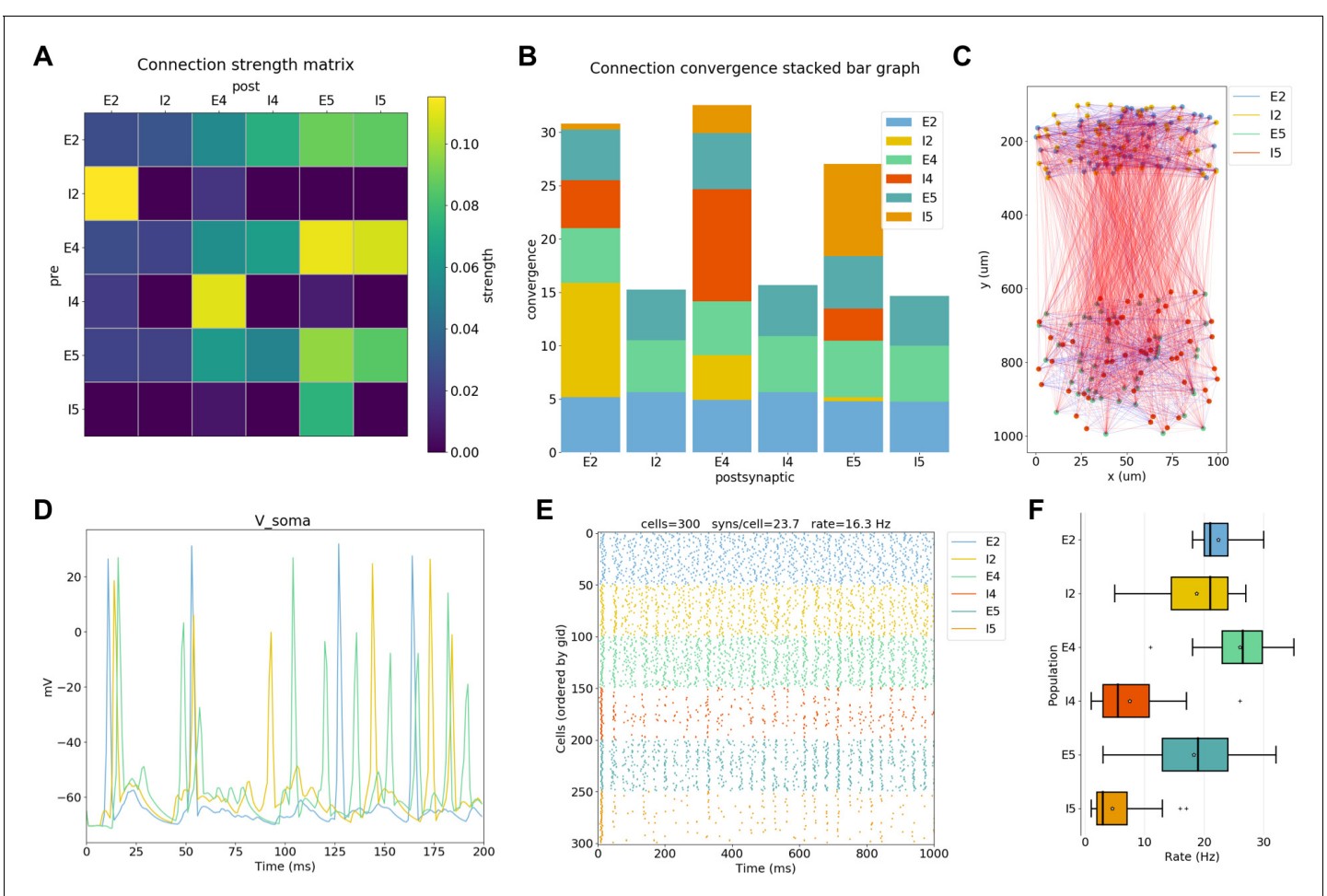

**Figure 5.** NetPyNE visualization and analysis plots for a simple three-layer network example. (A) Connectivity matrix, (B) stacked bar graph, (C) 2D graph representation of cells and connections, (D) voltage traces of three cells, (E) spike raster plot, (F) population firing rate statistics (boxplot).
DOI: https://doi.org/10.7554/eLife.44494.007

*2013*) and LFPsim (*Parasuram et al., 2016*) add-ons to NEURON. The LFP signal at each electrode is obtained by summing the extracellular potential contributed by each neuronal segment, calculated using the 'line source approximation' and assuming an Ohmic medium with conductivity (*Parasuram et al., 2016*; *Buzsáki et al., 2012*). The user can then plot the location of each electrode, together with the recorded LFP signal and its power spectral density and spectrogram (*Figure 6*). The ability to record and analyze LFPs facilitates reproducing experimental datasets that include this commonly used measure (*Buzsáki et al., 2012*).

## Data saving and exporting

NetPyNE permits saving and loading of all model components and results separately or in combination: high-level specifications, network instance, simulation configuration, simulation data, and simulation analysis results. Saving network instances enables subsequent loading of a specific saved network with all explicit cells and connections, without the need to re-generate these from the high-level connectivity rules. NetPyNE supports several standard file formats: pickle, JSON, MAT, and HDF5. The use of common file formats allows network structure and simulation results to be easily analyzed using other tools such as MATLAB or Python Pandas.

Network instances can also be exported to or imported from NeuroML (*Cannon et al., 2014*), a standard declarative format for neural models, and SONATA (https://github.com/AllenInstitute/sonata), a format standard for neural models proposed by the Blue Brain Project and Allen Institute for Brain Science. These formats are also supported by other simulation tools, so that models developed using NetPyNE can be exported, explored and simulated in other tools including Brian (*Goodman, 2008*), MOOSE (*Bower and Beeman, 2012*; *Ray and Bhalla, 2008*), PyNN (*Davison, 2008*), BioNet (*Gratiy et al., 2018*) or Open Source Brain (*Gleeson et al., 2018*). Similarly, simulations from these other tools can be imported into NetPyNE. This feature also enables any NetPyNE model to be visualized via the Open Source Brain portal, and permits a NeuroML model hosted on the portal to be parallelized across multiple cores (e.g. on HPC) using NetPyNE. Support for saving output simulation data to the standardized HDF5-based Neuroscience Simulation Data Format (NSDF) (*Ray et al., 2016*) is under active development.

Long simulations of large networks take a long time to run. Due to memory and disk constraints, it is not practical to save all state variables from all cells during a run, particularly when including signaling concentrations at many locations via the using the reaction-diffusion module. Therefore, NetPyNE includes the option of recreating single-cell activity in the context of spike inputs previously recorded from a network run. These follow-up simulations do not typically require an HPC

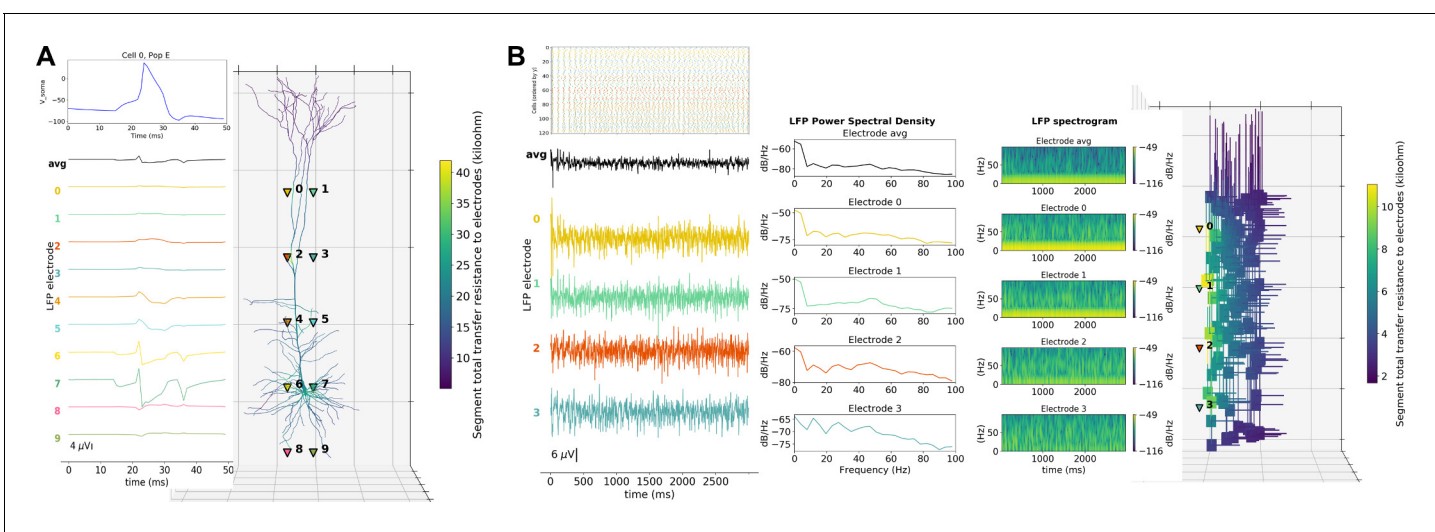

**Figure 6.** LFP recording and analysis. (**A**) LFP signals (left) from 10 extracellular recording electrodes located around a morphologically detailed cell (right) producing a single action potential (top-right). (**B**) LFP signals, PSDs and spectrograms (left and center) from four extracellular recording electrodes located at different depths of a network of 120 five-compartment neurons (right) producing oscillatory activity (top-left).

DOI: https://doi.org/10.7554/eLife.44494.008

since they are only running the single neuron. The user selects a time period, a cell number, and a set of state variables to record or graph.

## Parameter optimization and exploration via batch simulations

Parameter optimization involves finding sets of parameters that lead to a desired output in a model. This process is often required since both single neuron and network models include many under-constrained parameters that may fall within a known biological range of values. Network dynamics can be highly sensitive, with small parameter variations leading to large changes in network output. This then requires searching within complex multidimensional spaces to match experimental data, with degeneracy such that multiple parameter sets may produce matching activity patterns (*Edelman and Gally, 2001*; *Prinz et al., 2004*; *Neymotin et al., 2016b*). A related concept is that of parameter exploration. Once a model is tuned to reproduce biological features, it is common to explore individual parameters to understand their relation to particular model features, for example how synaptic weights affect network oscillations (*Neymotin et al., 2011*), or the effect of different pharmacological treatments on pathological symptoms (*Neymotin et al., 2016b*; *Knox et al., 2018*).

Many different approaches exist to perform parameter optimization and exploration. Manual tuning requires expertise and a great deal of patience (*Van Geit et al., 2008*; *Moles, 2003*). Therefore, NetPyNE provides built-in support for several automated methods that have been successfully applied to both single cell and network optimization: grid-search (*Achard and De Schutter, 2006*) and various types of evolutionary algorithms (EAs) (*Dura-Bernal et al., 2017*; *Neymotin et al., 2017*; *Carlson et al., 2014*; *Rumbell et al., 2016*; *Markram et al., 2015*; *Gouwens et al., 2018*). Grid search refers to evaluating combinations on a fixed set of values for a chosen set of parameters, resulting in gridded sampling of the multidimensional parameter space. EAs search parameter space more widely and are computationally efficient when handling complex, non-smooth, high-dimensional parameter spaces (*Moles, 2003*). They effectively follow the principles of biological evolution: here a population of models evolves by changing parameters in a way that emulates crossover events and mutation over generations until individuals reach a desired fitness level.

NetPyNE provides an automated parameter optimization and exploration framework specifically tailored to multiscale biophysically-detailed models. Our tool facilitates the multiple steps required: (1) parameterizing the model and selecting appropriate ranges of parameter values; (2) providing fitness functions; (3) customizing the optimization/exploration algorithm options; (4) running the batch simulations; and (5) managing and analyzing batch simulation parameters and outputs. To facilitate parameter selection and fitness function definitions, all the network specifications and simulation outputs are available to the user via the NetPyNE declarative data structure – from molecular concentrations and ionic channel conductances to long-range input firing rates. This frees the user from having to identify and access parameters or state variables at the NEURON simulator level.

Both parameter optimization and exploration involve running many instances of the network with different parameter values, and thus typically require parallelization. For these purposes, NetPyNE parallelization is implemented at two levels: (1) simulation level – cell computations distributed across nodes as described above; and (2) batch level – many simulations with different parameter values executed in parallel (*Dura-Bernal et al., 2017*). NetPyNE includes predefined execution setups to automatically run parallelized batch simulations on different environments: (1) multiprocessor local machines or servers via standard message passing interface (MPI) support; (2) the Neuroscience Gateway (NSG) online portal, which includes compressing the files and uploading a zip file via RESTful services; (3) HPC systems (supercomputers) that employ job queuing systems such as PBS Torque or SLURM (e.g. Google Cloud Computing HPCs). Users are able to select the most suitable environment setup and customize options if necessary, including any optimization algorithm metaparameters such as population size or mutation rate for EAs. A single high-level command will then take care of launching the batch simulations to optimize or to explore the model.

## Graphical user interface (GUI)

The GUI enables users to intuitively access NetPyNE functionality. It divides the workflow into two tabs: (1) network definition and (2) network exploration, simulation and analysis. From the first tab it is possible to define – or import from various formats – the high-level network parameters/rules and

simulation configuration (*Figure 2B*). Parameter specification is greatly facilitated by having clearly structured and labeled sets of parameters, graphics to represent different components, drop-down lists, autocomplete forms and automated suggestions. The GUI also includes an interactive Python console and full bidirectional synchronization with the underlying Python-based model – parameters changed via the Python console will be reflected in the GUI, and vice versa. In the second tab the user can interactively visualize the instantiated network in 3D, run parallel simulations and display all the available plots to analyze the network and simulation results. An example of a multiscale model visualized, simulated and analyzed using the GUI is shown in *Figure 7*. A description of this model was provided in the *Reaction-diffusion parameters* subsection.

The GUI is particularly useful for beginners, students or non-computational researchers, who can leverage it to rapidly build networks without advanced programming skills and without learning Net-PyNE's declarative syntax. From there, they can simulate and explore multiscale subcellular, cellular and network models with varying degrees of complexity, from integrate-and-fire up to large-scale simulations that require HPCs. The GUI is also useful for modelers, who can easily prototype new models graphically and later extend the model programmatically using automatically generated Python scripts. Finally, the GUI is useful – independently of expertise level – to explore and visualize existing models developed by oneself, developed by other users programmatically, or imported from other simulators. Understanding unfamiliar models becomes easier when users can navigate through all the high-level parameters in a structured manner and visualize the instantiated network structure, instead of just looking at the model definition source code (*McDougal et al., 2015*).

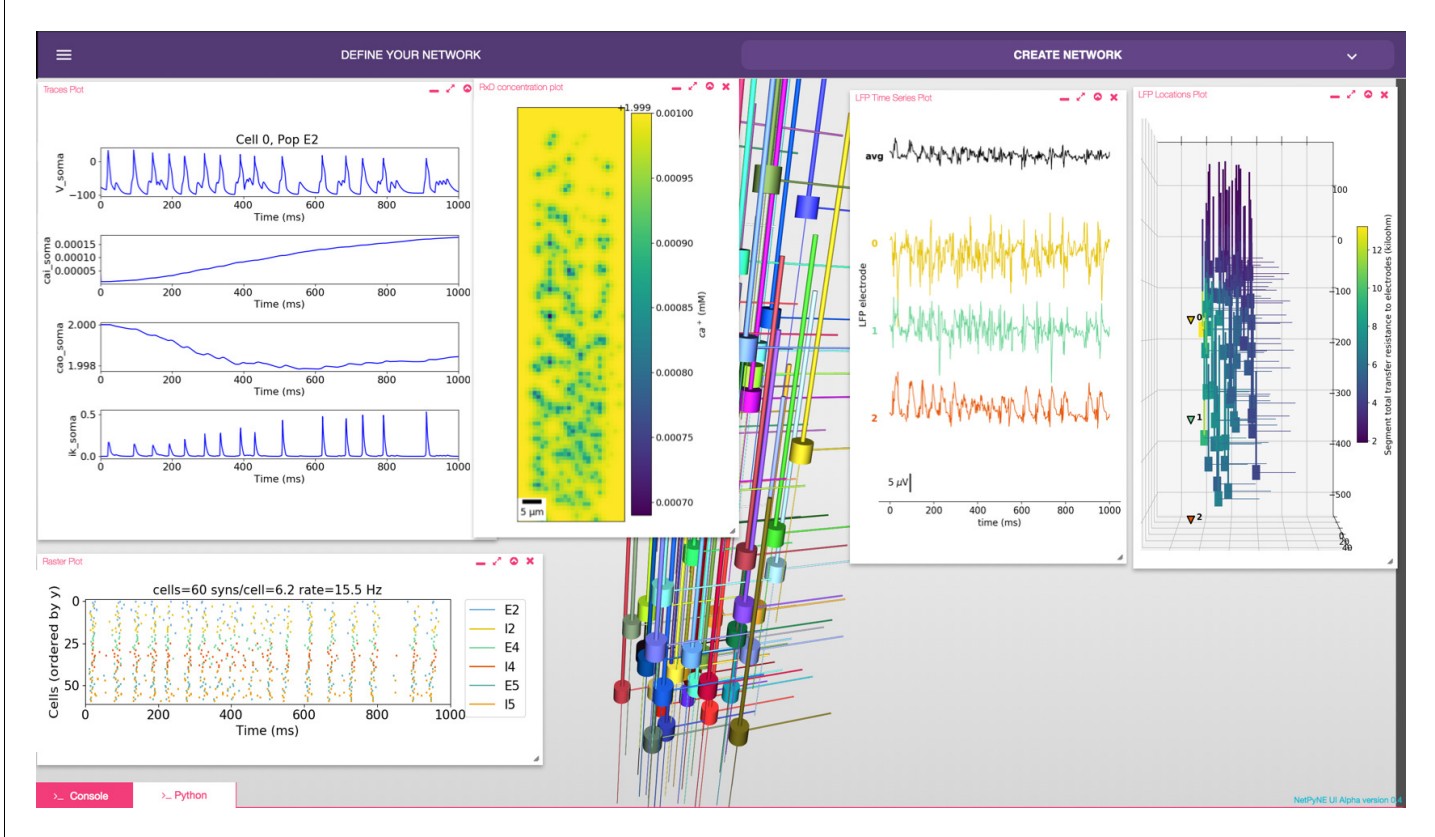

**Figure 7.** NetPyNE graphical user interface (GUI) showing a multiscale model. Background shows 3D representation of example network with 6 populations of multi-channel, multi-compartment neurons; results panels from left to right: singe-neuron traces (voltage, intracellular and extracellular calcium concentration, and potassium current); spike raster plot; extracellular potassium concentration; LFP signals recorded from three electrodes; and 3D location of the LFP electrodes within network.

DOI: https://doi.org/10.7554/eLife.44494.009

## Application examples

Our recent model of primary motor cortex (M1) microcircuits (*Dura-Bernal et al., 2018*; *Neymotin et al., 2016b*; *Neymotin et al., 2017*) constitutes an illustrative example where NetPyNE enabled the integration of complex experimental data at multiple scales: it simulates over 10,000 biophysically detailed neurons and 30 million synaptic connections. Neuron densities, classes, morphology and biophysics, and connectivity at the long-range, local and dendritic scale were derived from published experimental data (*Suter et al., 2013*; *Yamawaki et al., 2015*; *Yamawaki and Shepherd, 2015*; *Harris and Shepherd, 2015*; *Sheets et al., 2011*; *Weiler et al., 2008*; *Anderson et al., 2010*; *Yamawaki et al., 2015*; *Kiritani et al., 2012*; *Apicella et al., 2012*; *Hooks et al., 2013*; *Suter and Shepherd, 2015*). Results yielded insights into circuit information pathways, oscillatory coding mechanisms and the role of HCN in modulating corticospinal output (*Dura-Bernal et al., 2018*). A scaled down version (180 neurons) of the M1 model is illustrated in *Figure 8*.

Several models published in other languages have been converted to NetPyNE to increase their usability and flexibility. These include models of cortical circuits exploring EEG/MEG signals (https://hnn.brown.edu/) (*Jones et al., 2009*; *Neymotin et al., 2018*), interlaminar flow of activity (*Potjans and Diesmann, 2014*; *Romaro et al., 2018*) (*Figure 9A*) and epileptic activity (*Knox et al., 2018*) (*Figure 9B*); a dentate gyrus network (*Tejada et al., 2014*; *Rodriguez, 2018*) (*Figure 9C*); and CA1 microcircuits (*Cutsuridis et al., 2010*; *Tepper et al., 2018*) (*Figure 9D*). As a measure of how compact the NetPyNE model definition is, we compared the number of source code lines (excluding comments, blank lines, cell template files and mod files) of the original and NetPyNE implementations (see *Table 1*).

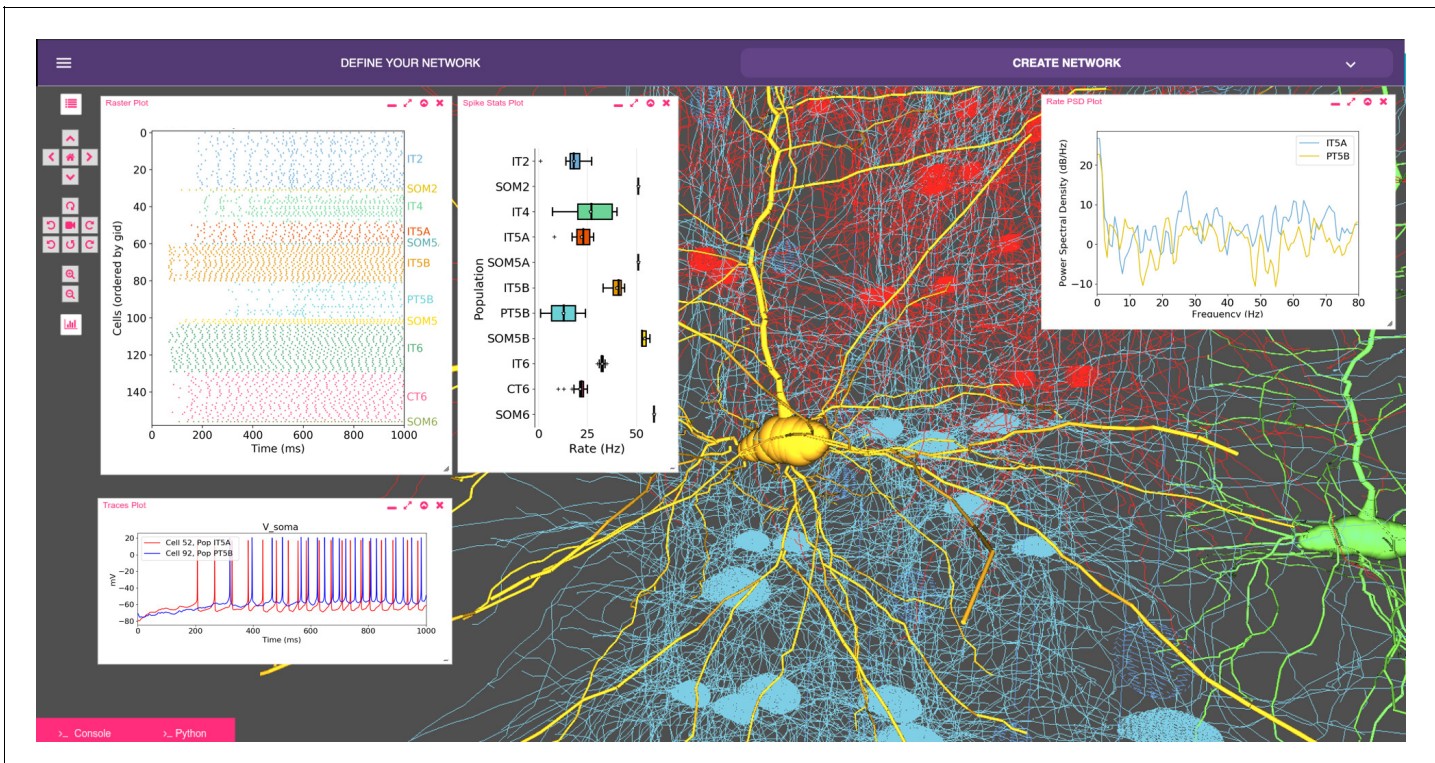

**Figure 8.** Model of M1 microcircuits developed using NetPyNE (scaled down version). NetPyNE GUI showing 3D representation of M1 network (background), spike raster plot and population firing rate statistics (top left), voltage traces (bottom left) and firing rate power spectral density (top right).

DOI: https://doi.org/10.7554/eLife.44494.010

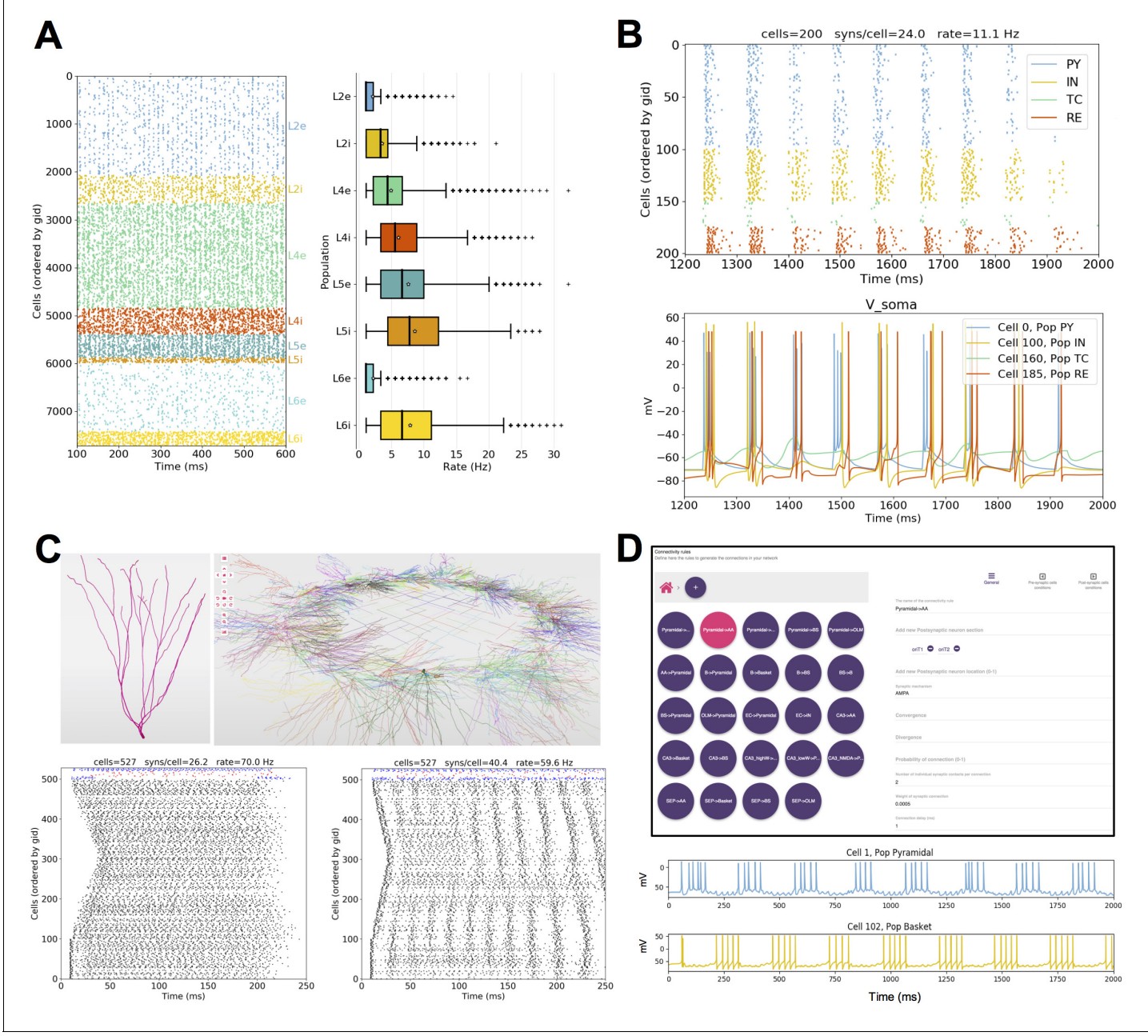

**Figure 9.** Published models converted to NetPyNE. All figures were generated using the NetPyNE version of the models. (A) Spike raster plot and boxplot statistics of the Potjans and Diesmann thalamocortical network originally implemented in NEST (*Potjans and Diesmann, 2014*; *Romaro et al., 2018*). (B) Spike raster plot and voltage traces of a thalamocortical network exhibiting epileptic activity originally implemented in NEURON/hoc (*Knox et al., 2018*). (C) 3D representation of the cell types and network topology, and spike raster plots of a dentate gyrus model originally implemented in NEURON/hoc (*Rodriguez, 2018*; *Tejada et al., 2014*). (D) Connectivity rules (top) and voltage traces of 2 cell types (bottom) in a hippocampal CA1 model originally implemented in NEURON/hoc (*Cutsuridis et al., 2010*; *Tepper et al., 2018*).
DOI: https://doi.org/10.7554/eLife.44494.011

## Discussion

NetPyNE is a high-level Python interface to the NEURON simulator that facilitates the definition, parallel simulation, optimization and analysis of data-driven brain circuit models. NetPyNE provides a systematic, standardized approach to biologically detailed multiscale modeling. Its broad scope offers users the option to evaluate neural dynamics from a variety of scale perspectives:

**Table 1.** Number of lines of code in the original models and the NetPyNE reimplementations.

| Model description (reference) | Original language | Original num lines | NetPyNE num lines |
| --- | --- | --- | --- |
| Dentate gyrus (*Tejada et al., 2014*) | NEURON/hoc | 1029 | 261 |
| CA1 microcircuits (*Cutsuridis et al., 2010*) | NEURON/hoc | 642 | 306 |
| Epilepsy in thalamocortex (*Knox et al., 2018*) | NEURON/hoc | 556 | 201 |
| EEG and MEG in cortex/HNN model (*Jones et al., 2009*) | NEURON/Python | 2288 | 924 |
| Motor cortex with RL (*Dura-Bernal et al., 2017*) | NEURON/Python | 1171 | 362 |
| Cortical microcircuits (*Potjans and Diesmann, 2014*) | PyNEST | 689 | 198 |

DOI: https://doi.org/10.7554/eLife.44494.012

for example (1) network simulation in context of the brain as an organ – that is with extracellular space included; (2) focus at the cellular level in the context of the network; (3) evaluate detailed spine and dendrite modeling in the context of the whole cell *and* the network, etc. Swapping focus back-and-forth across scales allows the investigator to understand scale integration in a way that cannot be done in the experimental preparation. In this way, multiscale modeling complements experimentation by combining and making interpretable previously incommensurable datasets (*Ferguson et al., 2017*). In silico models developed with NetPyNE serve as fully integrated testbeds that can be systematically probed to make testable predictions. Simulation can in some cases exceed the ability of physical experiments to build comprehension and develop novel theoretical constructs (*Markram et al., 2015*; *Dura-Bernal et al., 2016*; *Bezaire et al., 2016*; *MindScope et al., 2016*; *De Schutter and Steuber, 2009*).

To ensure accessibility to a wide range of researchers, including modelers, students and experimentalists, NetPyNE combines many modeling workflow features under a single framework with both a programmatic and graphical interface. The GUI provides an intuitive way to learn to use the tool and to explore all the different components and features interactively. Exporting the generated network to a Python script enables advanced users to extend the model programmatically.

## Multiscale specifications using a declarative language

By providing support for NEURON's intracellular and extracellular reaction-diffusion module (RxD) (*McDougal et al., 2013*; *Newton et al., 2018*), NetPyNE helps to couple molecular-level chemophysiology – historically neglected in computational neuroscience (*Bhalla, 2014*) – to classical electrophysiology at subcellular, cellular and network scales. RxD allows the user to specify and simulate the diffusion of molecules (e.g., calcium, potassium or IP3) intracellularly, subcellularly (by including organelles such as endoplasmic reticulum and mitochondria), and extracellularly in the context of signaling and enzymatic processing – for example metabolism, phosphorylation, buffering, and second messenger cascades. This relates the scale of molecular interactions with that of cells and networks (*Bhalla and Iyengar, 1999*).

NetPyNE rules allow users to not only define connections at the cell-to-cell level, but also to compactly express highly specific patterns of the subcellular distribution of synapses, for example depending on the neurite cortical depth or path distance from soma. Such distinct innervation patterns have been shown to depend on brain region, cell type and location; they are likely to subserve important information processing functions and have effects at multiple scales (*Komendantov and Ascoli, 2009*; *Kubota et al., 2015*; *Petreanu et al., 2009*; *Suter and Shepherd, 2015*). Some simulation tools (GENESIS [*Bower and Beeman, 2012*], MOOSE [*Ray and Bhalla, 2008*], PyNN [*Davison, 2008*] and neuroConstruct [*Gleeson et al., 2007*]) include basic dendritic level connectivity features, and others (BioNet [*Gratiy et al., 2018*]) allow for Python functions that describe arbitrarily complex synapse distribution and connectivity rules. However, NetPyNE is unique in facilitating the description of these synaptic distribution patterns via flexible high-level declarations that require no algorithmic programming.

NetPyNE's high-level language has advantages over procedural descriptions in that it provides a human-readable, declarative format, accompanied by a parallel graphical representation, making models easier to read, modify, share and reuse. Other simulation tools such as PyNN, NEST, Brian or BioNet include high-level specifications in the context of the underlying procedural language

used for all aspects of model instantiation, running and initial analysis. Procedural languages require ordering by the logic of execution rather than the logic of the conceptual model. Since the NetPyNE declarative format is order free, it can be cleanly organized by scale, by cell type, or by region at the discretion of the user. This declarative description is stored in standardized formats that can be readily translated into shareable data formats for use with other simulators. High-level specifications are translated into a network instance using previously tested and debugged implementations. Compared to creating these elements directly via procedural coding (in Python/NEURON), our approach reduces the chances of coding bugs, replicability issues and inefficiencies.

The trade-off is that users of a declarative language are constrained to express inputs according to the standardized formats provided, offering less initial flexibility compared to a procedural language. However, NetPyNE has been designed so that many fields are agglutinative, allowing multiple descriptors to be provided together to home in on particular subsets of cells, subcells or subnetworks, for example cells of a certain type within a given spatial region. Additionally, users can add procedural NEURON/Python code between the instantiation and simulation stages of NetPyNE in order to customize or add non-supported features to the model.

Developers of several applications and languages, including NeuroML, PyNN, SONATA and Net-PyNE, are working together to ensure interoperability between their different formats. NeuroML (*Cannon et al., 2014*) is a widely used model specification language for computational neuroscience, which can store instantiated networks through an explicit list of populations of cells and their connections, without higher level specification rules. We are collaborating with the NeuroML developers to incorporate high-level specifications similar to those used in NetPyNE, for example compact connectivity rules (see https://github.com/NeuroML/NeuroMLlite). The hope is that these compact network descriptions become a standard in the field so that they can be used to produce identical network instances across different simulators. To further promote standardization and interoperability, we and other groups working on large-scale networks together founded the INCF Special Interest Group on 'Standardized Representations of Network Structures' (https://www.incf.org/activities/standards-and-best-practices/incf-special-interest-groups/incf-sig-on-standardised). To facilitate the exchange of simulation output data, we are currently adding support for the Neuroscience Simulation Data Format (NSDF) (*Ray et al., 2016*), which was designed to store simulator-independent multiscale data using HDF5. Work is also in progress to extend NEURON's RxD partial support for reading and writing Systems Biology Markup Language (SBML), a standardized declarative format for computer models of biological processes (*Bulanova et al., 2014*). In the future, we aim to provide direct translation of SBML to NetPyNE's RxD declarative specifications.

## Integrated parameter optimization

A major challenge when building complex models is optimizing their many parameters within biological constraints to reproduce experimental results (*Van Geit et al., 2008*; *Moles, 2003*). Although there can be multiple solutions to observed dynamics, Marder and colleagues demonstrated that these are sparse in the space of possible solutions and that they correspond to physiologically reasonable ranges of the cell and synapse parameters, constrained but not precisely specified by experiment (*Golowasch et al., 2002*; *Prinz et al., 2003*).

Multiple tools are available to fit detailed single-cell models to electrophysiological data: BluePyOpt (*Van Geit et al., 2016*), Optimizer (*Friedrich et al., 2014*), pypet (*Meyer and Obermayer, 2016*) or NeuroTune (https://github.com/NeuralEnsemble/neurotune). However, these tools are limited to optimizing parameters and matching experimental data at the single-cell scale. NetPyNE provides a parameter optimization framework that covers the molecular, cellular and circuit scales, thus enabling and encouraging the exploration of interactions across scales. It also closely integrates with the simulator, rather than being a standalone optimizer, avoiding the need for an additional interface to map the data structures in both tools. This integration allows the user to select optimization parameters and specify fitness functions that reference the same data structures employed during model definition and analysis of simulation results. NetPyNE offers multiple optimization methods, including evolutionary algorithms, which are computationally efficient for handling the non-smooth, high-dimensional parameter spaces encountered in this domain (*Moles, 2003*; *Van Geit et al., 2008*; *Svensson et al., 2012*).

### Use of NetPyNE in education

In addition to the tool itself, we have developed detailed online documentation, step-by-step tutorials (http://netpyne.org), and example models. The code has been released as open source (https://github.com/Neurosim-lab/netpyne). Ongoing support is provided via a mailing list (with 50 subscribed users) and active Q and A forums (150 posts and over 5000 views in the first year): http://netpyne.org/mailing, http://netpyne.org/forum and http://netpyne.org/neuron-forum. Users have rapidly learned to build, simulate and explore models that illustrate fundamental neuroscience concepts, making NetPyNE a useful tool to train students. To disseminate the tool, we have also provided NetPyNE training at conference workshops and tutorials, summer schools and university courses. Several labs are beginning to use NetPyNE to train students and postdocs.

### Use of NetPyNE in research

Models being developed in NetPyNE cover a wide range of regions including thalamus, sensory and motor cortices (*Dura-Bernal et al., 2018*; *Neymotin et al., 2016b*), claustrum (*Lytton et al., 2017*), striatum, cerebellum and hippocampus. Application areas being explored include schizophrenia, epilepsy, transcranial magnetic stimulation (TMS), and electro- and magneto-encephalography (EEG/MEG) signals (*Sherman et al., 2016*). A full list of areas and applications is available at http://netpyne.org/models.

Tools such as NetPyNE that provide insights into multiscale interactions are particularly important for the understanding of brain disorders, which can involve interactions across spatial and temporal scale domains (*Lytton, 2008*; *Lytton et al., 2017*). Development of novel biomarkers, increased segregation of disease subtypes, new treatments, and personalized treatments, may benefit from integrating details of molecular, anatomical, functional and dynamic organization that have been previously demonstrated in isolation. Simulations and analyses developed in NetPyNE provide a way to link these scales, from the molecular processes of pharmacology, to cell biophysics, electrophysiology, neural dynamics, population oscillations, EEG/MEG signals and behavioral measures.

## Materials and methods

### Overview of tool components and workflow

NetPyNE is implemented as a Python package that acts as a high-level interface to the NEURON simulator. The package is divided into several subpackages, which roughly match the components depicted in the workflow diagram in *Figure 1*. The specs subpackage contains modules related to definition of high-level specifications. The `sim` subpackage contains modules related to running the simulation. It also serves as a shared container that encapsulates and provides easy access to the remaining subpackages, including methods to build the network or analyze the output, and the actual instantiated network and cell objects. From the user perspective, the basic modeling workflow is divided into three steps: defining the network parameters (populations, cell rules, connectivity rules, etc) inside an object of the class `specs.NetParams`; setting the simulation configuration options (run time, integration interval, recording option, etc) inside an object of the class `specs.SimConfig`; and passing these two objects to a wrapper function (`sim.createSimulateAnalyze()`) that takes care of creating the network, running the simulation and analyzing the output.

### Network instantiation

The following standard sequence of events are executed internally to instantiate a network from the high-level specifications in the `netParams` object: (1) create a Network object and add to it a set of `Population` and `Cell` objects based on `netParams.popParams` parameters; (2) set cell properties (morphology and biophysics) based on `cellParams` parameters (checking which cells match the conditions of each rule); (3) create molecular-level RxD objects based on `rxdParams` parameters; (4) add stimulation (IClamps, NetStims, *etc*) to the cells based on `stimSourceParams` and `stimTargetParams` parameters; and (5) create a set of connections based on `connParams` and `subConnParams` parameters (checking which pre- and post-synaptic cells match the connectivity rule conditions), with the synaptic parameters specified in `synMechParams`. After this process is completed all the resulting NEURON objects will be contained and easily accessible within a hierarchical Python structure (object `sim.net` of the `class Network`) as depicted in *Figure 4*.

The network building task is further complicated by the need to implement parallel NEURON simulations in an efficient and replicable manner, independent of the number of processors employed. Random number generators (RNGs) are used in several steps of the building process, including cell locations, connectivity properties and the spike times of input stimuli (e.g. NetStims). To ensure random independent streams that can be replicated deterministically when running on different numbers of cores we employed NEURON's Random123 RNG from the `h.Random` class. This versatile cryptographic-quality RNG (*Salmon et al., 2011*) is initialized using three seed values, which, in our case, will include a global seed value and two other values related to unique properties of the cells involved, for example for probabilistic connections, the gids of the pre- and post-synaptic cells.

To run NEURON parallel simulations NetPyNE employs a `pc` object of the class `h.ParallelContext()`, which is created when the `sim` object is first initialized. During the creation of the network, the cells are registered via the `pc` methods to enable exchange and recording of spikes across compute nodes. Prior to running the simulation, global variables, such as temperature or initial voltages, are initialized, and the recording of any traces (e.g. cell voltages) and LFP is set up by creating `h.Vector()` containers and calling the recording methods. After running the parallel simulation via `pc.solve()`, data (cells, connections, spike times, recorded traces, LFPs, *etc*) are gathered into the master node from all compute nodes using the `pc.py_alltoall()` method. Alternatively, distributed saving allows writing the output of each node to disk file and combines these files after the simulation has ended. After gathering, the built-in analysis functions have direct access to all the network and simulation output data via `sim.net.allCells` and `sim.allSimData`.

## Importing and exporting

NetPyNE enables import of existing cells in hoc or Python, including both templates/classes and instantiated cells. To achieve this, NetPyNE internally runs the hoc or Python cell model, extracts all the relevant cell parameters (morphology, mechanisms, point processes, synapses, *etc*) and stores them in the NetPyNE JSON-like format used for high-level specifications. The hoc or Python cell model is then completely removed from memory so later simulations are not affected.

Importing and exporting to other formats such as NeuroML or SONATA requires mapping the different model components across formats. To ensure validity of the conversion, we have compared simulation outputs from each tool, or converted back to the original format and compared to the original model. Tests on mappings between NetPyNE and NeuroML can be found at https://github.com/OpenSourceBrain/NetPyNEShowcase.

## Batch simulations

Exploring or fitting model parameters typically involves running many simulations with small variations in some parameters. NetPyNE facilitates this process by automatically modifying these parameters and running all the simulations based on a set of high-level instructions provided by the user. The two fitting approaches – grid search and evolutionary algorithms – both require similar set up. The user creates a `Batch` object that specifies the range of parameter values to be explored and the run configuration (e.g. use 48 cores on a cluster with SLURM workload manager). For evolutionary algorithms and optionally for grid search, the user provides a Python function that acts as the algorithm fitness function, which can include variables from the network and simulation output data (e.g. average firing rate of a population). The NetPyNE website includes documentation and examples on how to run the different types of batch simulations.

Once the batch configuration is completed, the user can call the `Batch.run()` method to trigger the execution of the batch simulations. Internally, NetPyNE iterates over the different parameter combinations. For each one, NetPyNE will (1) set the varying parameters in the simulation configuration (`SimConfig object`) and save it to file, (2) launch a job to run the NEURON simulation based on the run options provided by the user (*e.g.*, submit a SLURM job), (3) store the simulation output with a unique filename, and repeat for the next parameter set, or if using evolutionary algorithms, calculate the fitness values and the next generation of individuals (parameter sets).

To implement the evolutionary algorithm optimization we made use of the inspyred Python package (https://pythonhosted.org/inspyred/). Inspyred subroutines are customized to the neural environment using parameters and fitness values obtained from NetPyNE data structures, and running

parallel simulations under the NEURON environment either on multiprocessor machines via MPI or supercomputers via workload managers.

## Graphical user interface

The NetPyNE GUI is implemented on top of Geppetto (*Cantarelli et al., 2018*), an open-source platform that provides the infrastructure for building tools for visualizing neuroscience models and data and for managing simulations in a highly accessible way. The GUI is defined using JavaScript, React and HTML5. This offers a flexible and intuitive way to create advanced layouts while still enabling each of the elements of the interface to be synchronized with the Python model. The interactive Python backend is implemented as a Jupyter Notebook extension which provides direct communication with the Python kernel. This makes it possible to synchronize the data model underlying the GUI with a custom Python-based NetPyNE model. This functionality is at the heart of the GUI and means any change made to the NetPyNE model in the Python kernel is immediately reflected in the GUI and vice versa. The tool's GUI is available at https://github.com/Neurosim-lab/NetPyNE-UI and is under active development.

## Acknowledgements

This work was funded by grants from the NIH, NSF, NYS SCIRB, UK Welcome Trust and Australian Research Council, which are listed in detail below. We are thankful to all the contributors that have collaborated in the development of this open source tool via GitHub (https://github.com/Neurosim-lab/netpyne).

## Additional information

### Competing interests

Matteo Cantarelli, Facundo Rodriguez: is affiliated with MetaCell LLC. The author has no other competing interests to declare. Adrian Quintana: is affiliated with EyeSeeTea Ltd. The author has no other competing interests to declare. The other authors declare that no competing interests exist.

### Funding

| Funder | Grant reference number | Author |
|---|---|---|
| National Institute of Biomedical Imaging and Bioengineering | U01EB017695 | Salvador Dura-Bernal<br>Benjamin A Suter<br>Matteo Cantarelli<br>Adrian Quintana<br>Facundo Rodriguez<br>Samuel A Neymotin<br>Michael Hines<br>Gordon MG Shepherd<br>William W Lytton |
| New York State Department of Health | DOH01-C32250GG-3450000 | Salvador Dura-Bernal<br>Facundo Rodriguez<br>William W Lytton |
| Wellcome Trust | 101445 | Padraig Gleeson |
| National Institute on Deafness and Other Communication Disorders | R01DC012947-06A1 | Samuel A Neymotin |
| National Institute of Biomedical Imaging and Bioengineering | R01EB022903 | Salvador Dura-Bernal<br>Michael Hines<br>William W Lytton |
| National Institute of Mental Health | R01MH086638 | Robert A McDougal<br>Michael Hines<br>William W Lytton |
| Wellcome Trust | 212941 | Padraig Gleeson |

| National Institute of Biomedical Imaging and Bioengineering | R01EB022889 | Salvador Dura-Bernal Matteo Cantarelli Adrian Quintana Facundo Rodriguez Samuel A Neymotin Michael Hines |
| --- | --- | --- |
| Australian Research Council | DE140101375 | David J Kedziora Cliff C Kerr |

The funders had no role in study design, data collection and interpretation, or the decision to submit the work for publication.

## Author contributions

Salvador Dura-Bernal, Conceptualization, Data curation, Software, Formal analysis, Supervision, Funding acquisition, Validation, Investigation, Visualization, Methodology, Writing—original draft, Project administration, Writing—review and editing; Benjamin A Suter, Conceptualization, Data curation, Formal analysis, Validation, Investigation, Methodology, Writing—review and editing; Padraig Gleeson, Conceptualization, Software, Formal analysis, Validation, Investigation, Methodology, Writing—review and editing; Matteo Cantarelli, Conceptualization, Software, Validation, Investigation, Visualization, Methodology, Project administration, Writing—review and editing; Adrian Quintana, Facundo Rodriguez, Conceptualization, Software, Validation, Investigation, Visualization, Methodology, Writing—review and editing; David J Kedziora, Software, Formal analysis, Validation, Investigation, Methodology, Writing—review and editing; George L Chadderdon, Conceptualization, Methodology, Writing—review and editing; Cliff C Kerr, Samuel A Neymotin, Conceptualization, Software, Investigation, Methodology, Writing—review and editing; Robert A McDougal, Conceptualization, Software, Funding acquisition, Investigation, Methodology, Project administration, Writing—review and editing; Michael Hines, Conceptualization, Software, Formal analysis, Funding acquisition, Validation, Investigation, Methodology, Writing—review and editing; Gordon MG Shepherd, Conceptualization, Data curation, Funding acquisition, Investigation, Project administration, Writing—review and editing; William W Lytton, Conceptualization, Software, Supervision, Funding acquisition, Validation, Investigation, Visualization, Methodology, Writing—original draft, Project administration, Writing—review and editing

## Author ORCIDs

Salvador Dura-Bernal https://orcid.org/0000-0002-8561-5324
Benjamin A Suter https://orcid.org/0000-0002-9885-6936
Padraig Gleeson http://orcid.org/0000-0001-5963-8576
Matteo Cantarelli http://orcid.org/0000-0002-0054-226X
David J Kedziora http://orcid.org/0000-0002-0673-182X
Samuel A Neymotin http://orcid.org/0000-0003-3646-5195
Robert A McDougal http://orcid.org/0000-0001-6394-3127
Gordon MG Shepherd https://orcid.org/0000-0002-1455-8262
William W Lytton https://orcid.org/0000-0002-3727-2849

## Decision letter and Author response

Decision letter https://doi.org/10.7554/eLife.44494.015
Author response https://doi.org/10.7554/eLife.44494.016

# Additional files

## Supplementary files

• Transparent reporting form
DOI: https://doi.org/10.7554/eLife.44494.013

All data and models used in this work are publicly available from the following GitHub and ModelDB links:

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
