## [Decision Letter]

Thank you for submitting your article "NetPyNE: a tool for data-driven multiscale modeling of brain circuits" for consideration by *eLife*. Your article has been reviewed by three peer reviewers, one of whom is a member of our Board of Reviewing Editors, and the evaluation has been overseen by Ronald Calabrese as the Senior Editor. The following individuals involved in review of your submission have agreed to reveal their identity: David Sterratt (Reviewer #2); Daniel K Wójcik (Reviewer #3).

The reviewers have discussed the reviews with one another and the Reviewing Editor has drafted this decision to help you prepare a revised submission.

Summary:

This paper describes the NetPyNE project which provides a high-level declarative specification, python interface, and GUI for multiscale neuronal modeling. NetPyNE is a mature and powerful suite of software tools layered on the NEURON simulator, and the paper does a generally good job of an overview of this. The package itself, including documentation and examples, is its own best advocate.

The reviewers agreed that there were a number of points that need to beaddressed. These are all relatively straightforward items, and we expectthat the authors will be able to fix them in two months.

Essential revisions:

1) The installation process is not always straightforward. Some environmentsdon't seem to work right out of the box, and the instructions are not ascomplete as one would like.

2) There is some confusion about support for Python 2 vs Python 3. It is notalways clear which works best. The reviewers would be quite happy if it is just

Python 3, but in any event the Python version support for each of the reportedfeatures should be clearly stated.

3) Rainbow color maps (used in Figures 3 and 6) are not ideal. The authors should revisit this, and the reviewers make several suggestions.

In addition there are other relatively minor points made in the individualreviews, that the authors should readily be able to address.

*Reviewer #1:*

This paper describes the NetPyNE project which provides a high-level declarative specification, python interface, and GUI for multiscale neuronal modeling.

Overall, the paper describes a mature and powerful suite of software toolslayered on the NEURON simulator. This report discusses the model specification, the simulation environment especially for parallel simulations, the data analysis and visualization tools, the I/O formats, and the parameter tuningfeatures.

While the paper is clear and informative, the project is a very large one and the paper itself only provides an introduction to its capabilities. The project is its own best demonstration, which it does very well through documentation, examples, and tutorials, and above all through smooth functioning. Much of this review therefore examines the state of the software environment.

The project (github repository) is well used and is under continuousdevelopment. It seems to be used heavily and many publications mention NetPyNE.

The community has clearly been using it extensively, based on the large numberof github issues that are actively worked on.

The project seems to have borrowed good ideas from Brian2. Making connections is as easy as in Brian2 but expressions are built using python expressions rather than short Brian2 like expressions.

For integrating multiscale models, the implementation is on the lines ofrdesigneur, which is part of the MOOSE interface. The GUI (though in α)is pretty solid and intuitive to use.

The documentation is key to the project and is extensive. The quick start guide/tutorial is quite extensive and easy to start with. The reference document is also very solid, and it can be accessed from inside Python using the help() function.

Installation of NetPyNE was as easy as `pip install netpyne`. It requiresinstallation of NEURON and works with Python 3. Likewise the GUI, thoughin α, is readily installed through pip. This GUI is quite large (24MB source code) and has dependencies on more than 50 python packages. However the pip installation manages these. The system uses a modern GUI framework (html5+css) which is very portable and launches in the browser. It is slow to react but feature rich.

The authors also provide a docker image which may be a still easier way toget started, especially on different OS environments.

The example python scripts and tutorials ran without any problem, includingthe parallel examples.

I have only a few small critiques of the paper, particularly with regard tois rather sparse coverage of the RxD capabilities. RxD is mentioned severaltimes but other than some calcium dynamics its use is not well illustrated.

It isn't clear if there is currently a capability to read/write SBML, thoughthe NeuroML capability is well stated. In contrast, the network capabilitiesare very well fleshed out. Similarly, it isn't clear how ion channel kineticsmay be defined within this framework, or if they rely on pre-existing channeldefinitions.

*Reviewer #2:*

The article describes a new software tool, NetPyNE, which builds on the existing NEURON simulator. The tool is designed to facilitate building and sharing models of networks of neurons, simulating them (including on parallel architectures), simulating extracellular field potentials and ionic concentrations, and visualising the results. The main software tool is implemented as a Python module, which can be controlled via the Python prompt or Jupyter notebooks. There is also a GUI in development.

There are no new scientific findings as such, but this paper will be the standard citation for the NetPyNE tool, which I expect to be very well used, given how widely NEURON is already used. The software comes from the very well established group who already maintain the NEURON simulator, so the software is very likely to be maintained well into the future; NEURON and its predecessor CABLE have been around since around 1980.

I have tried using and downloading the software, confirming that the first tutorial example (http://www.netpyne.org/tutorial.html) works. I found that on my system (Scientific Linux 7) things did not quite work out of the box when I tried installing from source, but I would expect this from software that has not yet been tested widely in the field. From my previous experience of the NEURON maintainers, I expect that bug reports like mine (see below) will be dealt with quickly. Furthermore, the authors offer docker and virtual machine images, which should just work, though I didn't have time to try them.

The accuracy of the simulations will be largely determined by the underlying simulator (NEURON), but one important issue addressed by the software is that of starting simulations from random seeds, particularly in the case of parallel simulations.

There are software packages with overlapping functionality. For example LFPy (https://github.com/LFPy/LFPy) can compute EEG scalp surface potentials and MEG scalp surface magnetic fields, in addition to extracellular potentials. NEST (http://www.nest-simulator.org/) has been designed for simulating networks of relatively simple model neurons. However, I am not aware of a simulation package that covers all the areas that this one does.

There are some limitations to the connectivity structures possible to create, but this is a consequence of a declarative design, which does offer advantages over a programmatic generation of connectivity.

The paper is well-written, though I do have a small gripe with the rainbow colormap used in Figures 3 and 6, which I think could be designed to be more colorblind friendly. If viewed through Color Oracle (http://colororacle.org/index.html) with the grayscale option, it can be seen that the maximum intensity appears about halfway up the scale. Compare with a colorblind-friendly palette, such as Viridis: https://cran.r-project.org/web/packages/viridis/vignettes/intro-to-viridis.html

In summary, I recommend that this work is published in *eLife*, as it is important for enabling work in the field of network simulations, and helping with the reproducibility of simulations, which will contribute to increasing standards in the field of computational neuroscience.

*Reviewer #3:*

The present article advertises NetPyNE, a high level Python interface to NEURON simulator. It is a package of tools, many of which offer significant advantages to the modeler facilitating good programming practices which should lead to simpler, clearer, more reliable, better tested models. The major step is introduction of a new declarative language for definition of models which supports separation of model definition from its instantiation, simulation, and data analysis. The language facilitates use of simpler constructs at the level of NetPyNE as well as use of Neuron cell templates for more complex cell models. Multiple constructs facilitate building networks, different ways of distributing cells in space, synapses along the dendrites, and building connectivity. NetPyNE further facilitates multiscale modeling by its support to reaction-diffusion RxD part of Neuron and constructs supporting computation of extracellular potential.

Major features of model instantiation are a uniform and reproducible way of providing access within Python to every variable of the instantiated Neuron model and reliable and reproducible way of instantiating the models including random elements in parallel environments for different numbers of processors and cores. A range of tools for inspection of model and the results of simulation is provided to simplify these tasks, make them more accessible to less experienced users. The collection of analysis tools, while useful, I consider of lesser importance in view of the other features. Easy integrated access to extracellular potential is particularly useful. Both instantiated models and results of simulation can be saved in multiple formats.

NetPyNE provides tools for parameter optimization using genetic algorithms and grid search as well as tools for batch model running. It also provides a GUI which gives partial access to the package and may replace the standard Neuron interface in teaching on multiple levels, including experimental neuroscientists.

Table 2.9 indicates that the use of NetPyNE allows to shorten the length of the model definition around 3 times, further facilitating robustness of the model code.

I am convinced NetPyNE brings in very substantial added value to the existing simulator ecosystem. I will encourage its adoption among colleagues and in my lab. I am convinced its use will contribute to more robust and better tested neural models which in consequence will lead to many significant discoveries to better our understanding of the nervous system.

This article is mainly an advertisement for the software. The software is available openly at github and largely adequate documentation is provided online. The main features of the software are described adequately in the article, however, specifics are left to the documentation.

Main concern:http://netpyne.org/install.html declares that Python 2 or 3 are supported. https://github.com/MetaCell/NetPyNE-UI/wiki/Installing-NEURON-crxd-Version declares Python3 is in development. Which is the case? As this software is geared also towards less advanced users the installation instructions should be clear. I have a feeling they have not been tested by naive users.

What I expect is a clear information of how to install all tools relevant for the users of NetPyNE to use its all potential: multithreaded computation, RxD, GUI, etc. If the instructions are the same in Python 2 and 3 this should be made clear in the docs. If they differ, it should be clear. If only certain features work in one version of Python or Neuron, this should be acknowledged. I think this is much more important for the users than the information that it is possible to install NetPyNE without Neuron to translate models. This is an advanced feature which would not be used by many users, I suppose. The clear steps to installation should lead the user to an instance which at a minimum will allow him or her to run the tutorials, multithreaded computation, RxD, GUI. Assume no Neuron installed and recommend installation. Current instructions indicate --without-paranrn installation. Is this required for RxD? It is not clear.

Despite this concern I do recommend this article for publication after improving installation instructions and clarifying the issues of dependencies.

---

## [Author Response]

Essential revisions:1) The installation process is not always straightforward. Some environmentsdon't seem to work right out of the box, and the instructions are not ascomplete as one would like.

We have rewritten and reorganized the installation instructions to clarify the steps involved, and have fixed several issues identified by the reviewers. Further details are provided in the answers below.

2) There is some confusion about support for Python 2 vs Python 3. It is notalways clear which works best. The reviewers would be quite happy if it is justPython 3, but in any event the Python version support for each of the reportedfeatures should be clearly stated.

We have now clarified in the installation instructions that all NetPyNE features support both Python 2 and 3, except for the NetPyNE GUI, which only supports Python 3.

3) Rainbow color maps (used in Figures 3 and 6) are not ideal. The authors shouldrevisit this, and the reviewers make several suggestions.

We are thankful to the reviewers for bringing to our attention this issue, which we were not aware of. We have replaced the rainbow colormap in Figures 1, 3, 5, 6 and 7 with the Viridis colormap suggested by one of the reviewers. This colormap marks strength by intensity (equivalent to grayscale) and is colorblind friendly.

Reviewer #1:

[…] I have only a few small critiques of the paper, particularly with regard tois rather sparse coverage of the RxD capabilities. RxD is mentioned severaltimes but other than some calcium dynamics its use is not well illustrated.

It isn't clear if there is currently a capability to read/write SBML, thoughthe NeuroML capability is well stated. In contrast, the network capabilitiesare very well fleshed out. Similarly, it isn't clear how ion channel kineticsmay be defined within this framework, or if they rely on pre-existing channeldefinitions.

We agree with the reviewer that RxD was not adequately covered. We have created a separate subsection "Reaction-diffusion parameters" under the *"*High-level specifications", added two paragraphs describing the RxD components and how they are defined within NetPyNE. This includes a simple example to implement calcium buffering, which is also included in Figure 2A, and the description of a more detailed example involving IP3 and calcium signaling within the cytosol and endoplasmic reticulum, in a small cortical network.

To clarify kinetics definition we added the following: "RxD is now being extended to also permit definition of voltage-dependent or voltage- and ligand-dependent ion channels, and can also respond to synaptic events or interact with NMODL-defined mechanisms so as to affect membrane currents and membrane voltages."Experimental support for voltage-dependent ion channel kinetics has been added in a branch, and is expected to be available in NEURON 7.7, pending satisfactory completion of tests. See: *https://github.com/adamjhn/nrn/commit/c87ea34c848f47f3b6a1c90e95b3119e8117e170*

We added the following on SBML to the Discussion: *"*Work is also in progress to extend NEURON's RxD partial support for reading and writing Systems Biology Markup Language (SBML), a standardized declarative format for computer models of biological processes (Bulanova et al., 2014). We also aim to provide direct translation of SBML to NetPyNE's RxD declarative specifications."

Reviewer #2:[…] I have tried using and downloading the software, confirming that the first tutorial example (http://www.netpyne.org/tutorial.html) works. I found that on my system (Scientific Linux 7) things did not quite work out of the box when I tried installing from source, but I would expect this from software that has not yet been tested widely in the field. From my previous experience of the NEURON maintainers, I expect that bug reports like mine (see below) will be dealt with quickly.

We thank the reviewer for taking the time to install and try the software, and for identifying several issues. We have now fixed these issues and have provided more clear installation instructions. More details can be found in the responses below.

There are some limitations to the connectivity structures possible to create, but this is a consequence of a declarative design, which does offer advantages over a programmatic generation of connectivity.

We agree that the declarative approach has some limitations in terms of expressing connectivity. We have now clarified the text to better convey that connectivity patterns can also be defined via custom connectivity matrices; and that several parameters (e.g. probability, weight, delay, etc.) can be defined using arbitrarily-defined mathematical expressions that can include cell properties (e.g. location).

The paper is well-written, though I do have a small gripe with the rainbow colormap used in Figures 3 and 6, which I think could be designed to be more colorblind friendly. If viewed through Color Oracle (http://colororacle.org/index.html) with the grayscale option, it can be seen that the maximum intensity appears about halfway up the scale. Compare with a colorblind-friendly palette, such as Viridis: https://cran.r-project.org/web/packages/viridis/vignettes/intro-to-viridis.html

We have replaced all the colormaps in Figure 1, 3, 5, 6 and 7 with the Viridis colormap suggested by the reviewer.

Reviewer #3:[…] Main concernhttp://netpyne.org/install.html declares that Python 2 or 3 are supported. https://github.com/MetaCell/NetPyNE-UI/wiki/Installing-NEURON-crxd-Version declares Python3 is in development. Which is the case? As this software is geared also towards less advanced users the installation instructions should be clear. I have a feeling they have not been tested by naive users.What I expect is a clear information of how to install all tools relevant for the users of NetPyNE to use its all potential: multithreaded computation, RxD, GUI, etc. If the instructions are the same in Python 2 and 3 this should be made clear in the docs. If they differ, it should be clear. If only certain features work in one version of Python or Neuron, this should be acknowledged. I think this is much more important for the users than the information that it is possible to install NetPyNE without Neuron to translate models. This is an advanced feature which would not be used by many users, I suppose. The clear steps to installation should lead the user to an instance which at a minimum will allow him or her to run the tutorials, multithreaded computation, RxD, GUI. Assume no Neuron installed and recommend installation. Current instructions indicate --without-paranrn installation. Is this required for RxD? It is not clear.

We are thankful to the reviewer for taking the time to install and test the tool, and provide helpful feedback. We have updated the text and reorganized the installation instructions (http://www.netpyne.org/install.html) to make them more clear and to provide information on how to install the different required tools. We have also clarified that the basic NetPyNE package (without GUI) supports both Python 2 and Python 3, but the NetPyNE GUI only supports Python 3, and have provided instructions for each case. We made the installation with parallel support (*--with-paranrn*) the default option, and clarified that installing support for parallelization is optional, and that both options are compatible with RxD and the GUI. We have also removed comments about using NetPyNE without NEURON, and other unnecessary information.

We would also like to note that we have a working online version of NetPyNE (including the GUI, RxD and parallelization) which we plan to make permanently available this year as part of a collaboration with Open Source Brain. This will enable users to run the full tool online in a web browser without the need for any installation.